# A Bayesian-Symbolic Approach to Reasoning and Learning in Intuitive Physics

**Kai Xu**
University of Edinburgh
contact@xuk.ai

**Akash Srivastava**
MIT-IBM Watson AI Lab
akash.srivastava@ibm.com

**Dan Gutfreund**
MIT-IBM Watson AI Lab
dgutfre@us.ibm.com

**Felix A. Sosa**
Harvard University
fsosa@fas.harvard.edu

**Tomer Ullman**
Harvard University
tomerullman@gmail.com

**Joshua B. Tenenbaum**
Massachusetts Institute of Technology
jbt@mit.edu

**Charles Sutton**
University of Edinburgh & Google AI
c.sutton@ed.ac.uk

## Abstract

Humans can reason about intuitive physics in fully or partially observed environments even after being exposed to a very limited set of observations. This sample-efficient intuitive physical reasoning is considered a core domain of human common sense knowledge. One hypothesis to explain this remarkable capacity, posits that humans quickly learn approximations to the laws of physics that govern the dynamics of the environment. In this paper, we propose a Bayesian-symbolic framework (BSP) for physical reasoning and learning that is close to human-level sample-efficiency and accuracy. In BSP, the environment is represented by a top-down generative model of entities, which are assumed to interact with each other under unknown force laws over their latent and observed properties. BSP models each of these entities as random variables, and uses Bayesian inference to estimate their unknown properties. For learning the unknown forces, BSP leverages symbolic regression on a novel grammar of Newtonian physics in a bilevel optimization setup. These inference and regression steps are performed in an iterative manner using expectation-maximization, allowing BSP to simultaneously learn force laws while maintaining uncertainty over entity properties. We show that BSP is more sample-efficient compared to neural alternatives on controlled synthetic datasets, demonstrate BSP's applicability to real-world common sense scenes and study BSP's performance on tasks previously used to study human physical reasoning.[1]

## 1 Introduction

Imagine a ball rolling down a ramp. If asked to predict the trajectory of the ball, most of us will find it fairly easy to make a reasonable prediction. Not only that, simply by observing a single trajectory people can make reasonable guesses about the material and weight of the ball and the ramp. It is astonishing that while the exact answers to any of these prediction and reasoning tasks requires an in-depth knowledge of Newtonian mechanics and solving of some intricate equations, yet an average human can perform such tasks without any formal training in physics. Studies suggest that from early age humans come to understand physical interactions with very limited supervision, and can

---

[1]Source code as well as training and testing data can be accessed at https://bsp.xuk.ai/.

35th Conference on Neural Information Processing Systems (NeurIPS 2021).

efficiently reason and plan actions in common sense tasks, even in absence of complete information (Spelke, 2000; Battaglia et al., 2013). For example, with limited data, 4 or 5 years old children are capable of learning the physical laws behind magnetism (Bonawitz et al., 2019). Physical reasoning is considered a core domain of human common-sense knowledge (Spelke & Kinzler, 2007). Recent studies suggest that the ability to efficiently learn physical properties and interactions with limited supervision is driven by a noisy model of Newtonian dynamics, referred to as the *intuitive physics engine* (IPE; Bates et al., 2015; Gerstenberg et al., 2015; Sanborn et al., 2013; Lake et al., 2017; Battaglia et al., 2013). This has led to a surge in research aimed at developing agents with an IPE, or a model of the environment dynamics (Amos et al., 2018; Chang et al., 2016; Grzeszczuk & Animator, 1998; Fragkiadaki et al., 2015; Battaglia et al., 2016; Watters et al., 2017; Sanchez-Gonzalez et al., 2019; Ehrhardt et al., 2017; Kipf et al., 2018; Seo et al., 2019; Baradel et al., 2020). These efforts have created methods that either trade-off data-efficiency, by using deep neural networks (NNs), for high predictive accuracy (Breen et al., 2019; Battaglia et al., 2016; Sanchez-Gonzalez et al., 2019) or trade-off flexibility to learn from data for data-efficiency by using symbolic methods (Ullman et al., 2018; Smith et al., 2019; Sanborn et al., 2013; Bramley et al., 2018).

Inspired by the highly data-efficient ability of humans to learn and reason about their physical environment with incomplete information, we present Bayesian-symbolic physics (BSP), a Bayesian-symbolic model with an expectation-maximization (EM) algorithm that combines the sample efficiency of symbolic methods with the accuracy and generalization of data-driven approaches, using statistical inference of unobserved object properties and symbolic learning of physical force laws. In BSP, we model the evolution of the environment's dynamics over time as a generative program of entities interacting under Newtonian mechanics. As a probabilistic method, BSP treats the properties of entities, such as mass and charge, as random variables. Since Newtonian force laws are functions of these properties, in BSP we replace data-hungry NNs with symbolic regression (SR) to learn explicit force expressions, and then evolve them deterministically using equations of motion. A naive SR implementation here is not enough though due to two issues. One is that if it operates on a vanilla grammar that does not constrain the search space over force-laws, it can potentially have worse data-efficiency than a NN. Therefore, we introduce a *grammar of Newtonian physics* that leverages *dimensional analysis* to induce a physical unit system over the search space and impose physics-based constraints on the production rules. This prunes physically meaningless laws, therefore, drastically speeding up SR. Another issue is that the symbolic force expressions usually contain global constants, e.g. the gravitational constant, to learn, and common ways to deal with this challenge turn out to be inefficient especially in an EM setup. We tackle this challenge by using SR in a *bilevel optimization* framework in which a lower-level gradient based optimization step is used to optimize the constant. In short, our three main contributions are:

- We introduce a Bayesian-symbolic model for physical dynamics and an EM based algorithm, which combines approximate inference methods and SR, for maximum likelihood learning.
- We introduce a grammar of Newtonian physics that appropriately constrains SR for data-efficient learning, based on priors from dimensional analysis and physics-based constraints.
- Empirically, we show that BSP reaches human-like data-efficiency, often requiring just 1 to 5 synthetic scenes to learn the underlying force laws – much more data efficient than the closest neural alternatives. We then illustrate how BSP can discover physical laws from real-world common sense scenes from Wu et al. (2016). Finally, we study BSP on tasks previously used to study human physical reasoning in Ullman et al. (2018) and discuss the similarity and differences with human results.

## 2 Related work

Many symbolic and data driven models of learning and reasoning about physics can be broken down into smaller components that are either learned or fixed. In figure 1, we compare some of the closely-related recent work on physics learning. Starting on the right end, we have fully learned, deep NN approaches such as that used by Breen et al. (2019). This approach does not use any prior knowledge about physics, and learns to predict dynamics in a purely data-driven way. In the middle are hybrid models that introduce some prior knowledge about physical interactions or dynamics, in their NN-based prediction models. These include interaction networks (INs; Battaglia et al., 2016), ODE graph networks (OGNs), and Hamiltonian ODE graph networks (HOGNs; Sanchez-Gonzalez et al., 2019). Since these middle approaches use deep NNs, they tend to have very good predictive accuracy, yet poor sample complexity, requiring orders of magnitude more data to train than humans

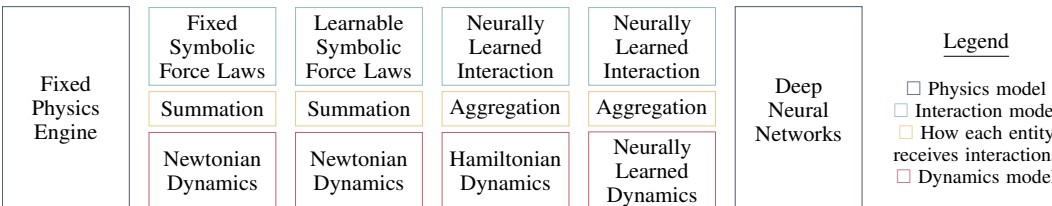

Figure 1: From left to right are rule-based to purely data-driven models of physics. Examples for each column are (1) Smith et al. (2019), (2) Ullman et al. (2018), (3) BSP (Ours), (4) (H)OGN Sanchez-Gonzalez et al. (2019), (5) IN Battaglia et al. (2016) and (6) Breen et al. (2019).

(Ullman et al., 2018; Battaglia et al., 2016; Sanchez-Gonzalez et al., 2019). On the other end of the spectrum (left) are fully symbolic, rule-based physics models and engines (Smith et al., 2019; Allen et al., 2019; Wu et al., 2015; Ullman et al., 2018). While these methods are suitable for reasoning tasks, they lack the flexibility of data-driven, learned models as they cannot generalize or adapt to changes in the environment that their fixed physics engine cannot simulate. For example, inference can fail on physically implausible scenes, and may require additional workarounds such as 'low probability events' outside of the dynamics (Smith et al., 2019).

Symbolic regression has been used for general physics learning in prior research, ranging from Schmidt & Lipson (2009)'s work on discovering force laws from experimental data, to the more recent work of Cranmer et al. (2020) on distilling symbolic forces from INs using genetic algorithms. More recently, Udrescu & Tegmark (2020) proposed AI Feynman, which recursively simplifies the SR problem using dimensional analysis and symmetries inferred by neural networks, to discover the underlying physics equations of the data. The focus of these types of work has been to discover underlying laws and equations based on *direct* input-output data. The focus of BSP, on the other hand, is on physics learning based on *indirect* signals from the environment; this is a task of interest in both intuitive physics studies with humans, and for human-like AI. Further, most symbolic approaches learn physics while assuming all properties in a system are known, which renders them inapplicable to environments with incomplete information. Some neural approaches focus on addressing such limitations in an end-to-end fashion (Zheng et al., 2018; Veerapaneni et al., 2020; Janner et al., 2019).

## 3 Bayesian-symbolic physics

BSP represents the physical environment using a generative model that evolves under Newtonian dynamics (section 3.1). In this model, physical laws are treated as learnable symbolic expressions and learned using symbolic regression and a specialized grammar of Newtonian physics that confines the search space and prevent the model from learning physically meaningless laws (section 3.2). BSP does not require all properties of the entities to be fully observed. It models these properties as latent random variables and infers them using Bayesian learning. To fit BSP on data, with incomplete information, we propose an EM algorithm that iterates between Bayesian inference of the latent properties, and SR which gives maximum likelihood estimation of the force expressions (section 3.3).

### 3.1 Generative model of the environment

In BSP's generative model, we represent each entity $i \in \{1 \ldots N\}$ by a vector of intrinsic physical properties $z^i$ (such as mass, charge, and shape), and a time dependent state vector $\mathbf{s}_t^i = (\mathbf{p}_t^i, \mathbf{v}_t^i)$ which describes the evolution of its position $\mathbf{p}_t^i \in \mathbb{R}^d$ and velocity $\mathbf{v}_t^i \in \mathbb{R}^d$ under Newtonian dynamics. Here, $d$ refer to the dimensionality of the environment, and is typically 2 or 3. Let $\{\tau^i\}_{i=1}^N$ be the set of observed trajectories from an environment with $N$ entities, where $\tau^i = \mathbf{p}_{1:T}^i := (\mathbf{p}_1^i, \ldots, \mathbf{p}_T^i)$. Then, together with the prior on $z$, for an observed trajectory data, $\mathcal{D}$, the generative model of BSP defines a joint probability distribution $p(\mathcal{D}, z; F)$ over $\mathcal{D}$ and latent properties $z$, given the force

Figure 2: Illustration of how the dimensional analysis and translation invariance priors help constrain the search space. Each box contains a subset of valid and illegal (stroked) sub-expressions.

function $F$.[2] The state transition of an entity in a Newtonian system depends on its properties and current state as well as its interaction with other entities. So, in BSP the force on entity $i$ at time $t$ is defined as $\mathbf{f}_t^i = \sum_{j=1}^{N} F(z^i, \mathbf{s}_t^i, z^j, \mathbf{s}_t^j)$, where $F(z^i, \mathbf{s}_t^i, z^j, \mathbf{s}_t^j)$ is the interaction force between entities $i$ and $j$. Then, the trajectory $\tau_i$ of entity $i$ is generated by a transition function $\mathbb{T}$ that consumes the current state and the resultant force to compute $\mathbf{s}_{t+1}^i = \mathbb{T}\left(\mathbf{s}_t^i, \mathbf{f}_t^i\right)$. Similar to Sanchez-Gonzalez et al. (2019), we use numerical integration to simulate the Newtonian dynamics inside $\mathbb{T}$. Specifically, we choose the Euler integrator and expand $\mathbb{T}$ as

$$\mathbf{a}_t^i = \mathbf{f}_t^i / m^i, \quad \mathbf{v}_{t+1}^i = \mathbf{v}_t^i + \mathbf{a}_t \Delta t, \quad \mathbf{p}_{t+1}^i = \mathbf{p}_t^i + \mathbf{v}_{t+1}^i \Delta t, \tag{1}$$

where $m^i$ is the mass of the recipient of the force $\mathbf{f}_t^i$ and $\Delta t$ is the step size of the Euler integrator. Finally, we add Gaussian noise to each trajectory $\{\tau^i\}_{i=1}^{N}$, that is, $\mathcal{D} := \{\tilde{\tau}^i\}_{i=1}^{N}$ where $\tilde{\tau}^i := (\tilde{\mathbf{p}}_1^i, \ldots, \tilde{\mathbf{p}}_T^i)$, $\tilde{\mathbf{p}}_t^i \sim \mathcal{N}(\mathbf{p}_t^i, \sigma^2)$ and $\sigma$ is the noise level. See appendix A.1 for the details of the complete generative process and illustrative examples.

## 3.2 A grammar of Newtonian physics

In order to attain good data efficiency, we choose to learn the pairwise force $F(z^i, \mathbf{s}^i, z^j, \mathbf{s}^j)$ between entities $i$ and $j$ using symbolic search. This approach can be inefficient if the search space of possible functions is too large, or inaccurate if the search space is too small. So, we constrain the function $F$ to be a member of a context-free language with a grammar $\mathcal{G}$, which we call *the grammar of Newtonian physics*. We design the grammar to be expressive enough to represent a large variety of potential force laws, while incorporating some simple, general constraints to improve the efficiency of symbolic search. Here we describe $\mathcal{G}$ informally; for the formal description, see figure 11 (appendix A.2).

We consider the following terminal nodes in $\mathcal{G}$: the masses $m_i, m_j$ of the entities, their friction coefficients $\mu_i, \mu_j$, shapes $s_i, s_j$, positions $\mathbf{p}_i, \mathbf{p}_j$, velocities $\mathbf{v}_i, \mathbf{v}_j$ the contact point $\mathbf{c}$ i.e. the position (if any) at which they touch, and finally a set of $K$ learnable constants $\{c_k\}_{k=1}^{K}$. In cases of no contact, $\mathbf{c}$ is set as the middle position of the two objects, i.e. $\mathbf{c} = (\mathbf{p}_i + \mathbf{p}_j)/2$. We include the operators: $(\cdot)^2$ (square), $+, -, \times, \div, \|\cdot\|_2$ (L2-norm), $\mathrm{normalise}(\cdot)$ and $\mathrm{project}(\cdot, \cdot)$, which projects a vector onto the unit ball.[3] The grammar also allows forces to be conditioned on a Boolean expression, in order to support *conditional forces* that only apply when a condition is true, e.g., when two objects collide. We provide $\mathcal{G}$ two primitive functions that encode the output of the perception system: $\mathrm{doesCollide}$ for collision detection and $\mathrm{isOn}$ to check if an entity is on a surface. These functions output integers 0 or 1. A rule in the grammar then allows a force expression to be multiplied by a conditional, so that BSP can learn expressions that represent when a conditional force should be applied.

Naively supporting all possible expressions for any combination of terminals would make SR highly inefficient, and even lead to *physically impossible* force laws. Therefore, we introduce two simple and general types of prior knowledge inspired by physics: dimensional analysis and translation invariance. Figure 2 shows examples of expressions that are excluded by each. First, inspired by dimensional analysis in natural sciences, where the relations between different units of measurement are tracked (Brescia, 2012), we built the concept of *units of measurement* into the nonterminals of $\mathcal{G}$. The units we consider are kilogram ($Kg$) for mass, meter ($Meter$) for distance, and meter per second ($MeterSec$) for speed. With this unit system in place, we only allow addition and subtraction of symbols with the *same* units, avoiding physically impossible sub-expressions such as $Kg - Meter$.[4]

---

[2]As physical dynamics are typically sensitive to *initial states*, we assume the noise-free initial states are given either as part of the data $\mathcal{D}$, or can be accurately estimated in a pre-processing step, e.g. by using consecutive positions, thus are omitted in the notation.

[3]In our work we consider maximally three forces to be learn in the same time, thus setting $K = 3$; more learnable constants and/or entity properties can be easily added to the grammar if needed.

[4]Constraining the grammar using units shares a similar spirit to type inference in type-directed synthesis (Feser et al., 2015; Osera & Zdancewic, 2015), which are crucial to improve search by avoiding illegal programs.

Importantly, this can lead to force laws with unit Newton ($N$).[5] Second, the grammar ensures that all force laws are *translation-invariant*, that is, independent on the choice of the origin of the reference frame. To do this, the grammar forbids the direct use of *absolute* positions $\mathbf{p}_i$, $\mathbf{p}_j$ and $\mathbf{c}$ and only allows their differences to be used expressions.[6]

Finally, some care is needed to ensure the grammar is unambiguous. For example, if we used a rule like $Coeff \rightarrow Coeff \times Coeff$, then the grammar could generate many expressions that redundantly represent the same function. This would make search much more expensive. Instead, we rewrite this rule in an equivalent right-branching way, e.g., $Coeff \rightarrow BaseCoeff \times Coeff$. This significantly reduces the search space without changing the expressivity of the grammar. Overall, although the grammar puts basic constraints on plausible physical laws, it is still expressive: there are more than *7 million* possible trees up to depth 6 while even the expression of universal gravitation has a depth of 7; the number of expressions up to depth 7 in $\mathcal{G}$ is intractable to count.

### 3.3 Learning algorithm

Following the EM approach, our learning method alternates between an E-step, where object property distributions are estimated given the current forces via approximate inference, and an M-step step, where forces are learned given object property distributions via SR (section 3.3.1). For the E-step, we consider two standard inference options: importance sampling (for any prior) and Hamiltonian Monte Carlo (for continuous priors only). Appendix A.3.1 discusses some details on applying them in BSP. Note appendix A.3 also provides all pseudo-code for algorithms introduced in this section.

**Implementation** In our work, we implement the generative models as probabilistic programs using the Turing probabilistic programming language (Ge et al., 2018) in Julia. As such the E-step is simply done by Turing's built-in samplers. For the M-step, we use the cross-entropy implementation from the ExprOptimization.jl package which allows users to define grammars with intuitive syntax.

#### 3.3.1 Symbolic regression with bilevel optimization for learnable constants

Symbolic regression (SR) is a function approximator that searches over a space of mathematical expressions defined by a context free grammar (CGF) (Koza, 1994). In our work, we use the cross-entropy method for SR. The method starts with a Probabilistic context-free grammars (PCFG) that assumes a uniform distribution over the production rules (PCFGs extend CFGs by assigning each production rule a probability). At each successive iteration, it samples $n$ trees (up to depth $d$) from the current PCFG, evaluates their fitness by a loss function $\mathcal{L}$, and uses the top-$k$ trees to fit a PCFG via maximum likelihood for the next iteration. This process returns the learned force law at the end of the training. More formally, to learn force laws, we need to find an expression $e \in L(\mathcal{G})$, where $L(\mathcal{G})$ is the language generated by $\mathcal{G}$, and values for the learnable constants $c := \{c_i\}_{k=1}^{3}$ that define the force function $\mathbf{f}_{e,c}$. The loss used by the cross-entropy method involves computing the log-likelihood of the generative model. As the observed trajectory is generated sequentially given an initial state, the computation of the log-likelihood term cannot be parallelized, and can be computationally expensive in practice. Therefore, following Battaglia et al. (2016) and Sanchez-Gonzalez et al. (2019), we use a *vectorized* version of the log-likelihood that basically performs simulation in each time stamp in parallel $LL(e, c; z, \mathcal{D}) = \sum_{i=1}^{N}\sum_{t=1}^{T-1} \log \mathcal{N}(\tilde{\mathbf{p}}_{t+1}^i; \mathbb{T}\left(\mathbf{s}_t^i, \mathbf{f}_{e,c,t}^i\right), \sigma)$ where $\mathbb{T}$ is expanded following equation 1 and $\tilde{\mathbf{s}}_t^i := (\tilde{\mathbf{p}}_t^i, \tilde{\mathbf{v}}_t^i)$. Clearly, $LL$ differs from its sequential corresponding, as the input for the integrator contains noise at each step. However, similar to previous work, we found it is not an issue when learning forces by regression.

In order to prevent overfitting by finding over-complex expressions, we add a regularization term – weighted log-probability under a uniform PCFG prior of $\mathcal{G}$ – to the negative log-likelihood; to arrive at our final loss per trajectory $\mathcal{L}(e, c; z, \mathcal{D}) = -LL(e, c; z, \mathcal{D}) + \lambda \log \mathcal{P}_0(e)$. Here $\mathcal{P}_0$ is the uniform PCFG of $\mathcal{G}$, and $\lambda$ is the hyper-parameter that controls the regularization. The loss for multiple trajectories is just a summation of $\mathcal{L}$ over individual trajectories. The continuous constants $c$ require care as they can take any value. To handle this, we use *bilevel optimization* (Dempe, 2002),

---

[5]When forming the unit $N$, constants can in fact has arbitrary units. But if there is any sub-expression like $Kg - Meter$, which is disallowed by our grammar, the final expression would not have any proper unit.

[6]This is consistent with how such variables are pre-processed in neural network approaches (Breen et al., 2019). Usually the mean of a pair of positions are subtracted from the pair to make them translation-invariant.

where the upper-level is the original symbolic regression problem, and the lower-level is an extra optimization for constants. This means we optimize the constants before computing the loss of each candidate tree within the cross-entropy iterations. The defined loss for each expression $e$ in SR is then $\mathcal{L}(e; z, \mathcal{D}) = \mathcal{L}(e, \arg\min_c \mathcal{L}(e, c); z, \mathcal{D})$. In BSP, we use the L-BFGS optimizer to solve the lower-level optimization.

Our way of handling learnable constants is related to other SR methods and bilevel optimization. Traditionally, constants are either randomly generated from a predefined, fixed integer set or a continuous interval, or for evolutionary algorithms, they can be mutated and combined during evolution to produce constants that fit better; such constants are often referred as *ephemeral constants* (Davidson et al., 2001). Compared to these methods, the benefit of our formulation is that the evaluation of each tree candidate depends on the symbolic form *only* as the constants are *optimized-away*, making the search more efficient. Note that although the literature has not explicitly considered our way of constant learning as bilevel optimization problem, similar strategies are also used in (Cerny et al., 2008; Kommenda et al., 2013; Quade et al., 2016). In contrast to recent use of bilevel optimization in meta-learning, e.g. (Finn et al., 2017), our method is simpler: As our upper-level optimization is gradient-free, we do not need to pass gradient from the lower-level to the upper-level.

## 4 Experiment: Learning force laws in fully observed environment

In this section, we evaluate the BSP in a data-limited setting when the properties are fully observed.

**Synthetic datasets (SYNTH).** We created three synthetic datasets for controlled evaluation: NBODY (n-body simulations with 4 bodies), BOUNCE (bouncing balls) and MAT (mats with friction); see figure 3 for an illustration. NBODY is populated by placing a heavy body with large mass and no velocity at $[0, 0]$, and three other bodies at random positions with random velocities such that,

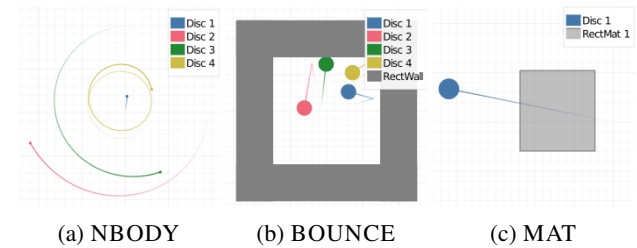

(a) NBODY     (b) BOUNCE     (c) MAT

Figure 3: Example scenes from SYNTH. Entities in gray are static.

they orbit the heavy body in the middle in the absence of the other two bodies. The gravitational constant is set such that the system is stable for the duration of the simulation. The ground truth force to learn is the gravitational force between bodies. BOUNCE is generated by simulating *elastic collisions* between balls in a box. The ground truth force to learn is the collision resolution force. MAT simulates friction-based interaction between discs and a mat. We populate this dataset by rolling discs over mats and applying a friction force when they come into contact. We randomized the initial states of the discs as well as the sizes, friction coefficients, and positions of the mats. The ground truth force to learn is the force of friction.

All scenes are simulated using a physics engine with a time-discretization of 0.02, for 50 frames. We generate 100 scenes per dataset, and hold-out 20 of them for testing. Appendix C.1 provides the ground truth force expressions used to generate each dataset under our grammar.

### 4.1 Data-efficiency: Symbolic vs neural

**Baselines** For the experiments in this section we use four different neural baselines: (i) A specialized instance of the OGN model (Sanchez-Gonzalez et al., 2019) that only outputs the partial derivative of the velocity variable, unlike the original model that also outputs the partial derivative of the position variable. This is because under Newtonian dynamics, the partial derivative of the position variable is simply the velocity. (ii) An Interaction Network (IN) (Battaglia et al., 2016) (iii) A multi-layer perceptron-based force model (MLP (Force)) that directly outputs the force, and (iv) A multi-layer perceptron-based position model (MLP (Position)) that outputs the next position. See appendix C.2 for details of the neural architecture, training and parameterization setup for all the baselines. Lastly, as a reference, we also include the performance of a zero-force baseline ($F_0$), which corresponds to

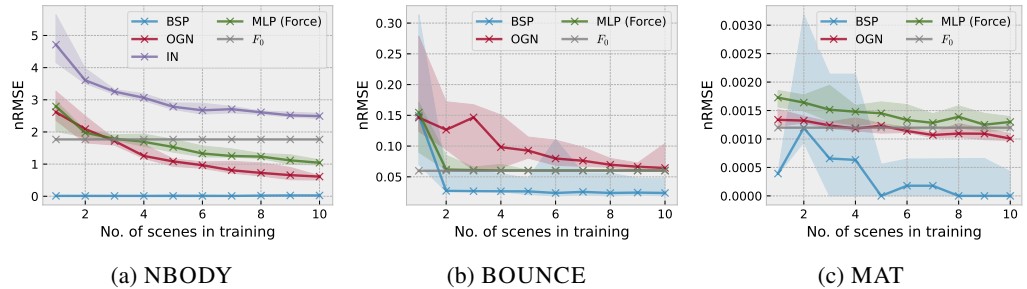

| (a) NBODY | (b) BOUNCE | (c) MAT |

Figure 4: Comparison of neural baselines and BSP, using predictive error on held out scenes given varying number of training scenes. Some baselines are not displayed due to very poor performance; see figure 16 in appendix C.4 for the version with all methods displayed.

the constant velocity baseline in (Battaglia et al., 2016). Note that all neural baselines as well as BSP are provided with symbolic representations for fair comparisons.

In order to compare the symbolic M-step of BSP against the neural baselines in terms of data-efficiency, we report the per-frame prediction accuracy on held-out datasets, as a function of the amount of training data. We use *noise-free* trajectories in this evaluation. Since the neural baselines cannot be trained if the properties are not fully observed, we provide all properties as observed data. For each dataset, we hold out 20 scenes for evaluation. We randomly shuffle the remaining 80 scenes, and use the first $k$ scenes to fit the models. Because an average person can perform physical learning task similar to the ones we use with fewer than 5 scenes (Ullman et al., 2018), we only vary $k$ from 1 to 10 in our experiments. We use the normalized root mean squared error (nRMSE), per frame per entity, between the predicted location of the entity and its actually location, as the performance metric. We repeat each of the experiments five times with different training set. These results are shown in figure 4, where the line plots are median values and the error bars are the 25% and 75% quantiles. Note that the ground truth force $F^*$ has an RMSE of 0 per frame. As can be seen, for most values of $k$ across of the three datasets, the symbolic M-step of BSP is more data-efficient than the neural baselines. The exceptions are in the MAT for $k = 1$ and in the MAT dataset for $k = 2, 3, 4$. This is likely due to specific bad local minima that may exist in the limited training data.

For NBODY and MAT, BSP can find the ground truth force function with 1 scene and 10 scenes respectively. BOUNCE is the only case where our method fails to find the true law within 10 scenes, we include the typical inferred force law in appendix C.3.1 as well the predicated trajectories of some selected scenes for inspection. In appendix C.3.2, we also demonstrate that this inferred force law closely approximates the true force law and so can generalize to other scenes.

For BOUNCE, the neural baselines cannot reach the performance of $F_0$ even after 10 scenes for training. This is a known issue with neural network approaches when learning collisions, as the inherently sparse nature of the collision interaction does not provide enough training signal (Battaglia et al., 2016). The rank of performance between neural baselines also supports our discussion around figure 1. Object-centeric modeling (OGN and IN) tends to have better performance than the rest by decomposing the transition into interaction and dynamics. Predefined dynamics with numerical integration (OGN and MLP force) have better performance than their correspondences with learned dynamics (IN and MLP position) by a notion of "force" (as the Euler integrator is used in this case).

In short, the symbolic regression with proper priors that constrain the search space in BSP leads to significantly better performance in terms of data efficiency across the three datasets studied. Together with the performance rank of neural models, our experiments shows how different levels and forms of prior help with data-efficient learning.

## 4.2 Ablation study of priors in the BSP grammar

To demonstrate the impact of the grammar of Newtonian physics on the overall data efficiency of BSP, we consider two ablations of our grammar: (i) $\mathcal{G}_{01}$, which is $\mathcal{G}$ without the dimensional analysis prior and $\mathcal{G}_{00}$, which is $\mathcal{G}$ without both priors in (i) and (ii); we also refer to $\mathcal{G}$ as $\mathcal{G}_{11}$ in this section. For reference, for a maximum depth of 5, $\mathcal{G}_{00}$ contains 8,593,200 expressions, $\mathcal{G}_{01}$ contains 7,935,408

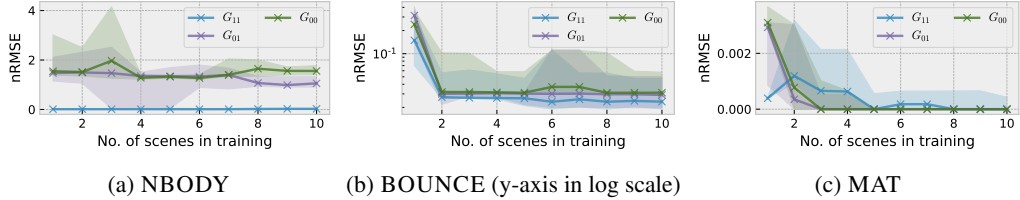

(a) NBODY           (b) BOUNCE (y-axis in log scale)         (c) MAT

Figure 5: Ablation study of priors using predictive error on held out scenes given varying number of training scenes. Comparison between $\mathcal{G}_{11}$ and $\mathcal{G}_{01}$ shows the effect of the dimensional analysis prior and comparison between $\mathcal{G}_{01}$ and $\mathcal{G}_{00}$ shows the effect of the translation invariance prior.

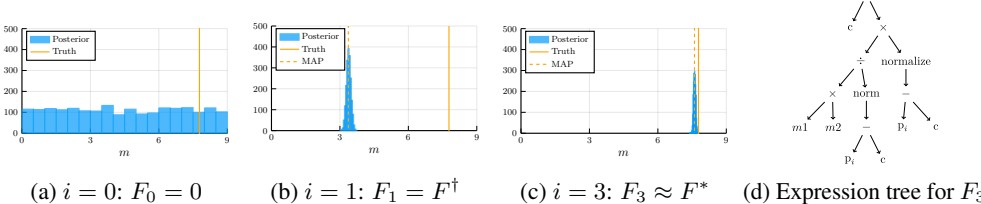

(a) $i = 0$: $F_0 = 0$      (b) $i = 1$: $F_1 = F^\dagger$      (c) $i = 3$: $F_3 \approx F^*$      (d) Expression tree for $F_3$

Figure 6: Results of the EM algorithm on NBODY. figure 6a to figure 6c shows the posterior of mass for Entity 1 in Scene 1 with the corresponding force function for different EM iteration $i$. In figure 6b, the force function $F^\dagger = 239.99 \frac{m_i m_j}{\|\mathbf{p}_i - \mathbf{p}_j\|_2} \frac{\mathbf{p}_i - \mathbf{p}_j}{\|\mathbf{p}_i - \mathbf{p}_j\|_2}$. The constant in figure 6d is $c = 2.04e3$.

expression and $\mathcal{G}_{11}$ contains 75,816. For a maximum depth of 6, $\mathcal{G}_{11}$ contains 771,120, and the number for other variants is intractable.

We repeat the experiment from section 4.1 for all three grammar variants, and report the results in figure 5. As can be seen, both priors in BSP's grammar contribute to data efficiency of its M-step while dimensional analysis has more impact. This is aligned with the analysis of the number of expressions per grammar above, showing that data efficiency improves as the number of possible expressions decreases. There is also a case in which the priors do not show advantages: MAT, in which the friction law is simple enough (the shallowest expression among all) thus easy to find even without priors.

## 5 Experiment: Learning force laws in partially observed environments

We now evaluate BSP's performance in environments with some unknown intrinsic entity properties.[7] We do not consider the neural baselines in this section, as they did not show competitive performance compared to BSP, even in the fully observed environment.

### 5.1 EM performance on SYNTH

We first demonstrate BSP's ability to jointly learn and reason about the environment by recovering the true force law when some properties are unobserved. As an illustrative example, we use three scenes from the NBODY dataset (with four entities per scene), such that if the true masses are given, the M-step can successfully learn the true force law. We assume that the mass of the heavy entity is known but the masses of other the three, lighter entities are unknown with a uniform prior $\mathcal{U}(0.02, 9)$. We use the EM algorithm to fit the same generative model that simulates the data using BSP. Figure 6 shows the posterior distribution over mass and the force function at initialization (figure 6a), middle (figure 6b), and convergence (figure 6c). In this run, after 3 iterations, our algorithm successfully recovers the true force function. We repeat this experiment ten times with randomly sampled scenes. For eight of them, BSP successfully recovers the true force law. Appendix D.1.1 provides another demonstrative example on MAT.

For a quantitative analysis, we run the EM algorithm on each of the three scenarios from SYNTH with 5 random scenes repeated 5 times. In table 1 we report nRMSE on a fixed testing set of 20

---

[7]It is worth to mention that there is an identifiability issue for joint reasoning-learning tasks with limited data, which we elaborate in appendix A.4.

| | NBODY | | | BOUNCE | | | MAT | | |
|---|---|---|---|---|---|---|---|---|---|
| | Median | 25% Q | 75% Q | Median | 25% Q | 75% Q | Median | 25% Q | 75% Q |
| BSP | 8.05e-1 | 7.83e-1 | 1.07e0 | 3.16e-2 | 3.11e-2 | 3.94e-2 | 5.39e-4 | 3.58e-4 | 7.63e-4 |
| $F_0$ | | 1.77e0 | | | 5.98e-2 | | | 1.20e-3 | |

Table 1: Test predictive performance (nRMSE) on partially observed SYNTH scenarios (using 5 random scenes for training and from 5 different runs). Note BSP consistently beats the constant baseline.

scenes. Notice that the performance has a large variance due to the fact that not all randomly selected 5-scene subsets provide enough training signal. In all three scenarios, BSP consistently outperforms the zero force baseline and can successfully recover the true force law in some random subsets.

## 5.2 Real world data: Physics 101

While SYNTH benchmark is interesting, a strength of our approach is the ability to generalize to real world data. To demonstrate this, we use the PHYS101 dataset (Wu et al., 2016), a dataset of real world physical scenes. We consider two scenarios, FALL and SPRING (shown in figure 7). As BSP works on symbolic inputs, we pre-process raw videos using standard tracking algorithms from OpenCV to extract observations in numerical form; see the appendix for pre-processing details.

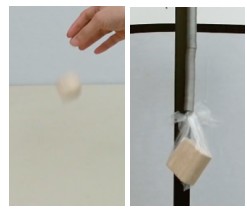

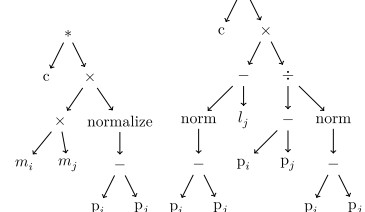

| | Mode | 25% Q | 75% Q |
|---|---|---|---|
| BSP | 4.30e-2 | 3.45e-2 | 5.09e-2 |
| $F_0$ | 5.05e-2 | 4.61e-2 | 5.49e-2 |
| BSP | 9.22e-3 | 8.25e-3 | 9.87e-3 |
| $F_0$ | 2.19e-2 | 2.17e-2 | 2.21e-2 |

Figure 7: Example frames for FALL (left) and SPRING (right)

Figure 8: Learned force expression for FALL (left) and SPRING (right)

Table 2: Test predictive performance (nRMSE) for FALL (top) and SPRING (bottom)

**FALL** We first train BSP on a single scene from FALL, where an object is dropped on a table. A typical force expression that BSP discovers is $c \times m_i \times m_j \times \text{normalize}(\mathbf{p}_i - \mathbf{p}_j)$, as shown in figure 8, suggesting that BSP is able to learn the correct form of gravitational force law. Here, the direction of $\mathbf{p}_i - \mathbf{p}_j$ points towards the table and $m_j$ is the mass of the table, which together with $c$, serves as the constant $g$ in $F_g = m_i g$. In another solution frequently found by BSP, it learns the direction as $\text{normalize}(\mathbf{v}_i)$ as the velocity is always downwards. BSP can also learn global forces directly if constant vectors $[1, 0]$ and $[0, 1]$ are provided, which is done in the next section.

**SPRING** After learning the gravity from FALL, in SPRING, assuming that the original length of the spring is known, we train BSP on a single scene to evaluate if it can learn the Hooke's law $F = kx$. Here $k$ is the tensor coefficient, and $x$ is the displacement of the spring. An example force law that BSP learns in this case is $c \times (\text{norm}(\mathbf{p}_i - \mathbf{p}_j) - l_j) \times (\mathbf{p}_i - \mathbf{p}_j) \div \text{norm}(\mathbf{p}_i - \mathbf{p}_j)$, as shown in figure 8, clearly suggesting that BSP can learn Hooke's law.

Finally, to quantitatively evaluate BSP's performance on PHYS101, we select a fixed set of 4 scenes from each scenario and use 2 for training and 2 for testing, repeating for all the permutations of the 4 scenes. The aggregated test performance in terms of normalized RMSE is given in table 2. BSP only slightly outperforms compared to the zero force baseline on the FALL. This is because the FALL contains only limited number of frames ($< 10$) and the trajectories do not diverge too far away from what zero force would predict. For SPRING, BSP outperforms the baseline significantly, closely estimating with ground truth time period of the harmonic motion. We also provide qualitative evaluation via a time-series plot of the change of the block's vertical position with time in the appendix.

## 5.3 Does BSP perform similarly to humans?

In this section, we compare BSP's performance against humans' on the experiment done in (Ullman et al., 2018). For this purpose, we use the ULLMAN dataset from this study, which consists of 60 videos in which a set of discs interact with each other and mats within a bounded area, as exemplified

in figure 9. While similar to SYNTH, ULLMAN has a lot more diversity in the scenes (stimulus) and reasoning tasks. The force laws in ULLMAN are similar to those in SYNTH but they have different constants and the scenes are generated from a completely different simulator. In the original experiment, each participant is presented with 5 videos. Each of the videos is from a different "world", such that the object properties (for each color) and force laws are different in every video. For each video, the participant is asked 13 multiple choice questions related to the mass of discs ("Mass"), roughness of mats ("Friction") and types of global ("Global") and pairwise forces ("Pairwise"). For example, "How massive are red objects?" where the options to choose from are "Light", "Medium" and "Heavy". Please refer to appendix D.3 for the complete set of questions and options.

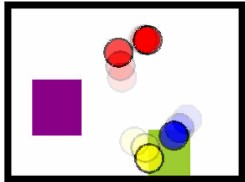

|          | Human | BSP  | Chance |
|----------|-------|------|--------|
| Mass     | 43%   | 40%  | 33%    |
| Friction | 44%   | 39%  | 33%    |
| Global   | 68%   | 55%  | 20%    |
| Pairwise | 62%   | 50%  | 33%    |

Figure 9: Example from ULLMAN     Table 3: Accuracy per question category

To be consistent with the setup in (Ullman et al., 2018), we assume that the friction and collision forces are known apriori. Thus the goal is to apply BSP on a **single** scene and infer the properties by learning the expressions for the residual global and pairwise force. The properties to infer are the mass for the discs, the friction coefficients for the mats and a latent property $q$ that controls the pairwise interaction. To accommodate for the global force, we added two constant vectors $[1, 0]$ and $[0, 1]$ to the grammar; properties related to the known forces are also removed from the grammar. In comparison to models studied in Ullman et al. (2018), BSP aims to learn the force expressions explicitly, rather than inferring binary variables to turn on/off predefined force components. Appendix D.3 provides details on the learning tasks and setup of BSP. We perform 3 runs of BSP on each of the 60 scenes and use the learning results to answer the same set of 13 questions presented to participants in (Ullman et al., 2018).

Table 3 summarizes the accuracy for humans and BSP on the four question categories. As it can be seen, BSP's performance is worse than that of humans' but convincingly better than chance. Considering the difficulty of inferring 9 properties and learning the targeted force law using only 1 scene, this may not be surprising. There are intriguing similarities between the answers given by BSP and human participants. Both display the same relative order of accuracy across question types "Global" > "Pairwise" > "Friction" > "Mass" while BSP's performance is still inferior to humans'. We hypothesize that humans may have much prior experience with similar physical scenes to answer these questions or they may answer these questions in a different way than explicitly learning the forces. Fully addressing the similarity and difference between BSP and humans requires more analysis that is out of the scope of this paper.

## 6    Limitation and Future Work

BSP relies on two main assumptions. First, we assume it has access to the grammar of Newtonian physics. However, how this knowledge is acquired, e.g. through evolution, is not addressed. Second, BSP works on symbolic inputs and assumes perceptual modules are given. This requires extra pre-processing steps when applying BSP to perceptual data, e.g. videos from PHYS101 and ULLMAN. BSP's performance also depends on the quality of such pre-processing step. It will be interesting to address these two limitations by formulating a computational framework in which the grammar and the perceptual modules are learned, either in separate phases or jointly with symbolic force learning.

## 7    Conclusion

We present BSP, a Bayesian approach to learning symbolic physics which, to our knowledge, is the first to combine symbolic learning of physical force laws and statistical learning of unobserved attributes. Our work enables data-efficient symbolic physics learning from partially-observed trajectory data and paves the way for using learnable IPEs in intuitive physics by providing a computational framework to study how humans' iterative reasoning-learning is mentally performed.

## Acknowledgment

We thank David D. Cox, John Cohn, Masataro Asai, Cole Hurwitz, Seungwook Han and all the reviewers for their helpful feedback and discussions. This work is supported by the DARPA Machine Common Sense (MCS) program.

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
