## A  Technical details

### A.1  The complete generative process

Section 3 describes the top-down generative model piece by piece. To improve the clarity the presentation, we provide the complete generative process of the observation given force function $F$, which corresponds to the E-step in our method, as a probabilistic program in algorithm 1. In this probabilistic program, we use the keyword ASSUME and OBSERVE for sampling latent variables and observations separately, following the notations from Wood et al. (2014). We also provide an example of the generative process of a three-body problem in a more intuitive manner in figure 10.

### A.2  Grammar for Newtonian physical laws: The complete form

The complete grammar following section 3.2 is given in figure 11.

---

**Algorithm 1** Complete generative process given force laws

                                                              ▷ Sample latent variables

1: **for** $i = 1, \ldots, N$ **do**
2:     ASSUME $z^i$ from prior for entity $i$
3:     **if** initial state is not given **then** ASSUME $\mathbf{p}_0^i$ from prior for entity $i$ ASSUME $\mathbf{v}_0^i$ from prior for entity $i$
4:     **end if**
5:     Set $\mathbf{s}_0^i = (\mathbf{p}_0^i, \mathbf{v}_0^i)$
6: **end for**
7: **for** $t = 1, \ldots, T$ **do**
8:     **for** $i = 1, \ldots, N$ **do**                                     ▷ Compute force and acceleration
9:         **for** $j = 1, \ldots, N$ **do**
10:            Compute $\mathbf{f}_t^{i,j} = F(z^i, \mathbf{s}_{t-1}^i, z^j, \mathbf{s}_{t-1}^i)$
11:         **end for**
12:         Compute $\mathbf{f}_t^i = \sum_{j=1}^N \mathbf{f}_t^{i,j}$
13:         Compute $\mathbf{a}_t^i = \mathbf{f}_t^i / m^i$                                   ▷ Euler's integration
14:         Update $\mathbf{v}_t^i = \mathbf{v}_{t-1}^i + \mathbf{a}_t \Delta t$
15:         Update $\mathbf{p}_t^i = \mathbf{p}_{t-1}^i + \mathbf{v}_t^i \Delta t$
16:         Set $\mathbf{s}_t^i = (\mathbf{p}_t^i, \mathbf{v}_t^i)$                                   ▷ Sample observations
17:         OBSERVE $\tilde{\mathbf{p}}_t^i$ from $\mathcal{N}(\mathbf{p}_t^i, \sigma^2)$
18:     **end for**
19: **end for**

---

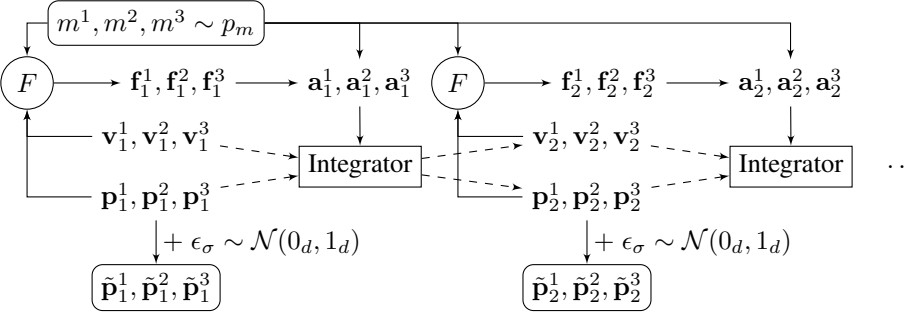

Figure 10: The generation of an observed trajectory: a three-body example with unknown mass. Circles are the learnable force function, rectangles are fixed functions, rounded rectangles are random variables and others are deterministic variables.

## A.3 Algorithmic description for the learning method

We provide a complete description of EM algorithm in algorithm 2.

We provide a complete description of the cross-entropy method with learnable constants in algorithm 3.

### A.3.1 Reasoning about unknown properties

**Importance sampling** In cases where the prior distribution is discrete, the inference is done by importance sampling (IS) which produces a set of weighted samples. As the number of samples to accurately estimate the marginal log-likelihood in the M-step can be large and this would induce a large computational cost in the M-step, IS is followed by a re-sampling step to select only a small set of $k$ weighted samples in the M-step $\{(\omega_1, z_1), \ldots, (\omega_k, z_k)\}$, where the weights are re-normalized.

**Hamiltonian Monte Carlo** Since for a fixed $F$, the generative model in BSP is end-to-end, piece-wise differentiable with respect to properties, we can use Hamiltonian Monte Carlo (HMC; Duane et al., 1987; Neal, 2011) for inference. In order to draw $k$ samples from the posterior *robustly* in the E-step, we first run $k + k'$ independent HMC chains by the no-U-turn sampler (NUTS; Hoffman

$$Constant \rightarrow c_1 \mid c_2 \mid c_3$$
$$Unitless \rightarrow \mu_1 \mid \mu_2$$
$$\mid \mu_1 - \mu_2 \mid \mu_1 + \mu_2$$
$$Kg \rightarrow m_i \mid m_j$$
$$\mid m_i - m_j \mid m_j + m_i$$
$$KgSq \rightarrow m_i \times m_j \mid (Kg)^2$$
$$MeterVec \rightarrow \mathbf{p}_i - \mathbf{p}_j$$
$$\mid \mathbf{p}_i - \mathbf{c} \mid \mathbf{p}_j - \mathbf{c}$$
$$MeterSecVec \rightarrow \mathbf{v}_i \mid \mathbf{v}_j \mid \mathbf{v}_i - \mathbf{v}_j$$
$$Meter \rightarrow \|MeterVec\|_2 \mid l_j$$
$$MeterSq \rightarrow (Meter)^2$$
$$MeterSec \rightarrow \|MeterSecVec\|_2$$
$$MeterSecSq \rightarrow (MeterSec)^2$$

$$TransInvVec \rightarrow MeterVec \mid MeterSecVec$$
$$UnitlessVec \rightarrow \text{normalize}(TransInvVec) \mid MeterVec \div Meter$$
$$\mid MeterSecVec \div MeterSec$$
$$Meter \rightarrow \text{project}(MeterVec, UnitlessVec)$$
$$MeterSec \rightarrow \text{project}(MeterSecVec, UnitlessVec)$$
$$BaseCoeff \rightarrow Unitless \mid Kg \mid KgSq \mid KgSq \div Kg \mid Meter$$
$$\mid MeterSq \mid Meter - Meter \mid Meter + Meter$$
$$\mid MeterSec \mid MeterSecSq + MeterSecSq$$
$$\mid MeterSecSq \mid MeterSecSq - MeterSecSq$$
$$Coeff \rightarrow BaseCoeff \mid BaseCoeff \times BaseCoeff$$
$$\mid BaseCoeff \div BaseCoeff$$
$$BaseForce \rightarrow Constant \times Coeff \times UnitlessVec$$
$$Bool \rightarrow \text{isOn}(\mathbf{p}_i, s_i, \mathbf{p}_j, s_j) \mid \text{doesCollide}(\mathbf{p}_i, s_i, \mathbf{p}_j, s_j)$$
$$Force \rightarrow BaseForce \mid BaseForce \times Bool \mid Force + BaseForce$$

Figure 11: A grammar of Newtonian physical laws

---

**Algorithm 2** Expectation–maximization for Bayesian-symbolic physics

**Input**: Dataset $\mathcal{D}$, grammar $\mathcal{G}$, number of EM iterations $m$, sample size $k$ and extra chains $k'$ in E-step, number of repeats in M-step $r$

**Output**: A force function $F$ and $k$ samples of latent properties $\{z_1, \ldots, z_k\}$

1: Initialize the force function $F_0$ as constantly zero
2: **for** $i = 1, \ldots, m$ **do**
3:     Get $k$ weighted posterior samples $\{(\omega_1, z_1), \ldots, (\omega_k, z_k)\}$ by IS or HMC          ▷ E-step
4:     Define current loss function $\mathcal{L}_i(e, c) = \sum_{i=1}^{k} \omega_i \mathcal{L}(e, c; z_i, \mathcal{D})$          ▷ M-step starts
5:     Get candidates $\mathbb{C} = \{(t_1^*, c_1^*), \ldots, (t_r^*, c_r^*)\}$ by algorithm 3 with $\mathcal{L}_i$ for $r$ repetitions
6:     Find $(t^*, c^*)$ from $\mathbb{C}$ with the best loss and set $F_i = \text{getF}(t^*, c^*, \mathcal{G})$          ▷ Update force
7: **end forreturn** $F = F_m$ and $\{z_1, \ldots, z_k\} \sim p(z \mid \mathcal{D}; F_m)$

---

& Gelman, 2014) for a reasonably large number of iterations, where $k'$ is a hyper-parameter. After this, we remove $k'$ chains with the smallest effective sample size (ESS). This reduces the chance of using samples from chains that mixed poorly or got stuck in bad region due to random initialization. Finally, we pick the last sample from each chain as the samples returned by the E-step $\{z_1, \ldots, z_k\}$. To be consistent with the samples from IS, we also assign equal weights $\omega_i = 1/k$ to all samples.

### A.4 Identifiability in reasoning and learning tasks

It is worth mentioning the fact that reasoning and learning tasks which BSP targets are not necessarily *identifiable*, especially when data is very limited or when force laws and object properties are jointly learned. When the data is limited, a certain level of *diversity of attribute values* in the data has to be provided so that their impact on the force law will be observed. For example, consider a dataset with multiple scenes of a 2-body simulation with the same 2 entities and random initialization of position and velocities. In this setup, no matter how many scenes are given, the actual gravitational force is not identifiable because the product of mass is a constant for all scenes. This can be resolved by introducing more entities in the same scene, or more scenes with entities that have different attributes. In the case of joint reasoning and learning, the interplay between attribute units and learnable constants in the force law could potentially create ambiguity. For example, if a force law acts on an attribute linearly, the learning algorithm is free to scale up the constant in the force law and scale down the attribute value accordingly to reach the same results. This can actually be seen by the fact that constants in force laws have their own units, e.g. the gravitational constant $G$ has a unit of $m^3 kg^{-1} s^{-2}$. Scaling the constant and the attribute accordingly can be seen as a unit change. Such ambiguity between properties and force laws is also the reason why one might not want to be Bayesian on force law, because there would be a mode switching problem in posterior sampling.

**Algorithm 3** Cross-entropy method with learnable constants
___
**Input**: Grammar $\mathcal{G}$ with learnable constants $c$, loss function $\mathcal{L}$, total population number $n$, selected population number $k$, number of iterations $m$ and maximum tree depth $d$
**Output**: An expression tree $e^*$ with optimized constants $c^*$
 1: initialize a PCFG $\mathcal{P}_0$ for $\mathcal{G}$ uniformaly
 2: **for** $i = 1, \ldots, m$ **do**
 3:     Initialize an empty candidate set $\mathbb{C}$
 4:     **for** $j = 1, \ldots, n$ **do**
 5:         Sample an expression $e_j \sim \mathcal{P}_{i-1}, e_{i-1}$ with a maximum depth of $d$
 6:         Solve $c_j^* = \arg\min_c \mathcal{L}(e_j, c)$ by L-BFGS           $\triangleright$ Lower-level optimization
 7:         Compute the loss of the sampled tree $\ell_j = \mathcal{L}(e_j, c_j^*)$ and add $(e_j, \ell_j)$ to $\mathbb{C}$
 8:     **end for**
 9:     **if** $i < m$ **then**
10:         Fit a PCFG $\mathcal{P}_i$ on trees from $\mathbb{C}$ with the top-$k$ fitness via maximum-likelihood
11:     **end if**
12: **end for return** the best expression tree $e^*$ from $\mathbb{C}$ and the corresponding constant as $c^*$
___

# B Reproducibility

Source code as well as training and testing data can be accessed at `https://bsp.xuk.ai/`.

# C Experimental Details for Section 4

All experiments are performed on CPUs using two servers. One has Intel(R) Xeon(R) CPU E5-2620 v3 @ 2.40GHz and the other has Intel(R) Xeon(R) CPU E5-2620 v4 @ 2.10GHz. The two servers has 24 + 32 = 56 cores in total to help run experiments in parallel.

## C.1 Ground truth forces

The symbolic trees of ground truth forces that are used to generate the datasets that are used in section 4 are given in figure 12.

## C.2 Neural baselines

We now describe the neural baselines. Notation-wise, we use $d_{\text{in}}$ to denote the total dimension of properties and state (position and velocity) for each entity, and use $d_{\text{out}}$ for the dimension of position/velocity/force dimension (2 in our case). The corresponding implementation can be found in `src/network.jl` in our source code, and the hyper-parameters (network sizes and training) can be found in `scripts/runexp.jl`, which we also summarise below.

**OGN** For the OGN baseline, we use a MLP of $d_{\text{in}} \to 100 \to 50$ as the node model and a MLP of $(50 + 50) \to 100 \to 100 \to 100 \to d_{\text{out}}$ as the edge model; the activation function is the rectified linear unit (ReLU) for both models. For training, we use the ADAM optimizer (Kingma & Ba, 2014) with a learning rate of $2\mathrm{e}{-3}$ for 2,000 epochs.

In addition, for OGN, we found that we need provide additional prior knowledge on how forces are related to the mass and acceleration by parameterzing them as $F_e(\cdot) = m\, a_\theta(\cdot)$, where $\theta$ is NN parameters, otherwise they fail to learn. This parameterization is fact consistent with (Sanchez-Gonzalez et al., 2019) in which NNs output partial derivatives of the Hamiltonian system.

**IN** The node model for IN is same as that of OGN. The edge model for IN is same as that of OGN except the output dimension of the last layer is 50. There is an extra network for transition, a MLP of shape $N \times 50 \to 100 \to 100 \to 100 \to 2 \times N \times d_{\text{out}}$ (with ReLU activations), that takes the concatenation of embeddings for $N$ entities and outputs the change of next state (position and velocity) of the whole system. The training uses the ADAM optimizer with a learning rate of $1\mathrm{e}{-3}$ for 400 epochs.

**MLP force** The MLP (force) baselines has a neural network that inputs the states of a pair of entities and outputs the force from one of them applies to the other. The network is a MLP of shape $2 \times d_{\text{in}} \rightarrow 100 \rightarrow 100 \rightarrow 100 \rightarrow d_{\text{out}}$ (with ReLU activations). The training uses the ADAM optimizer with a learning rate of $1\text{e}{-}3$ for 2,000 epochs.

**MLP position** The MLP (force) baselines has a neural network that inputs the state and properties of the whole system (as the concatenation of $N$ entities) and outputs the change of next state (position and velocity) of the whole system. The network is a MLP of shape $N \times d_{\text{in}} \rightarrow 100 \rightarrow 100 \rightarrow 100 \rightarrow 2 \times N \times d_{\text{out}}$ (with ReLU activations). The training uses the ADAM optimizer with a learning rate of $1\text{e}{-}3$ for 400 epochs.

### C.3 A close look at the approximated bounce law

#### C.3.1 The learned bounce law

As mentioned in section 4 and discussed in section C.3.2, the only case in which BSP fails to infer the true law (within 10 scenes) is of special interest and requires further inspection. A typical approximate law learned in section 4 is shown in figure 13; see section C.3.2 for discussion on how this law differs from the true one. To highlight, there are basically two mismatches between the true law and the learned law. First, there is no projection operation that correctly calculates the effect of speed. Second, the mass-based coefficient is missing. To assist inspection, we also provide some visualizations in figure 14 using initial conditions from the training set for inspection. The corresponding animations can be found in the supplementary material (the `suppl/bounce_inspection` folder in our repository).

#### C.3.2 Generalization in new scenes

It is worth checking how the laws learned in section 4 generalize to new scenes beyond the training data. In cases where the true law is successfully recovered, the expression will generalize to novel scenes undoubtedly. Therefore, it is more interesting to inspect the generalization ability of an approximate law, that is a law which is not completely equivalent to the true law but is close. The emerged law for the BOUNCE dataset is such an example as mentioned earlier. It has an expression of $F^{\dagger} = c \, \|\mathbf{v}_i - \mathbf{v}_j\|_2 \, \frac{\mathbf{p}_i - \mathbf{c}}{\|\mathbf{p}_i - \mathbf{c}\|_2}$ doesCollide$(\mathbf{p}_i, s_i, \mathbf{p}_j, s_j)$; see figure 13 in appendix C.3.1 for the actual tree. Although it is not identical to the true law, it is still a good approximation: it takes into accounts the velocity difference into consideration and finds the correct force direction. We now consider applying this law to a completely new scene: a vertical-view world where the gravity is pointing in downward direction. Figure 15 shows the predicted trajectory with true and the approximate law with two different initial conditions. As it can be seen, the approximate law successfully generalize this novel world. For the first condition, the projection is very close to the true one, while for the second condition, the concept of bounce is also correctly transferred. The corresponding animations for these plots can also be found in the supplementary material for further inspections (the `suppl/generalization` folder in our repository).

### C.4 Figure 4 with all methods displayed

Figure 4 omits some poor results for better visualization. The corresponding plots with all methods displayed are shown in figure 16.

## D Experimental details for Section 5

### D.1 SYNTH

**Hyper-parameters** We refer readers to `scripts/runexp.jl` of our source code for hyper-parameters used in the quantitative experiments (table 1). For the rest, in the E-step, we use $k = 3$ and $k' = 2$ and the hyper-parameters for NUTS are: 150 adaptation steps, 150 HMC iterations, a maximum tree depth of 4 and a target acceptance ratio of 0.75. In the M-step, we repeat $r = 2$ runs and the hyper-parameters for the cross-entropy method are: 800 total populations, 400 selected populations, 4 iterations and a maximum depth of 10. The weighting parameter for the PCFG prior are 1 for the NBODY and BOUNCE datasets and $1\text{e}{-}4$ for the MAT dataset.

### D.1.1 An extra demonstration of EM on MAT

As another example, we use five scenes from the (noisy) MAT dataset. We assume that the only unknown is the friction coefficient of the mat with a truncated Gaussian prior $\mathcal{T}\text{runcated}(\mathcal{N}(\mu_0, 2^2), 0, 5)$ (truncated between 0 and 5), where $\mu_0$ is the true coefficient, Note that the variance $2^2$ is large enough to be uncertain, justifying a fair choice of the prior. Similarly, we use the EM algorithm to fit the same generative model that simulates the data using BSP. figure 17 shows the posterior distribution over mass and the force function at initialization (17a), middle (17b) and convergence (17c) of the algorithm. Compared the expression at convergence with the true law, the algorithm learns $\mathbf{v}_i - \mathbf{v}_j$ instead of $\mathbf{v}_i$ as the mat velocity is zero, i.e. $\mathbf{v}_j = 0$, in all scenes,

## D.2 PHYS101

**Pre-processing of videos**  We use an open-source implementation of standard tracking algorithms from OpenCV to track the entities. The code is available at `https://github.com/bikz05/object-tracker/`. To use the tracker, we manually select a bounding box of the entity of interest and run the tracking algorithm. For FALL, it is done for the falling object; for SPRING, this is done for both the hanging object and the spring joint. For FALL, we found that the tracking algorithms can fail if the object is too fast. In such cases, there are only limited frames ($< 10$) to track thus we manually annotate these frames to get the trajectory of the falling object. The processed data can be found in `data/phys101/processed/` of the supplementary material.

**Qualitative evaluation**  To qualitatively see how well this learned force from SPRING performs at prediction, we also show that how the vertical coordination of the object position changes over time in figure 18. As it can be seen, the learned force produces prediction that matches the periodicity quite well with some small deviation from the amplitude.

## D.3 ULLMAN

**Pre-processing of visual stimulus**  We preprocess the videos from ULLMAN by derendering the objects to symbolic forms, i.e. position trajectories of all entities. This is done by template matching of the discs and mats. We manually crop the video frames to obtain templates for discs with three different colors and mats with three different colors, and match the location of each of them for each frame. The code for this preprocessing step can be found in `scripts/preprocess-ullman.py` of the source code and the processed data can be found in `data/ullman/processed/` of the supplementary material.

**On reverse-engineering the collision and friction forces from stimulus**  As the ULLMAN data is generated by an unknown simulator, the ground truth forces are not directly accessible. Therefore, we need to "reverse engineer" these forces so that we can provide them to BSP aprior, which is consistent with the setup of human study in Ullman et al. (2018). We assume the ground truth forces for friction and collision have their pre-defined expressions, similar to those used for the simulator for SYNTH. However, each of these expressions also contains a constant that is unknown for the actual simulation of the ULLMAN data. To this end, we use World 1 from the ULLMAN data to fit these constants because World 1 contains only friction and collision. For the rest of experiments of BSP, we assume these "reverse engineered" forces are given and BSP only needs to learn the residual, as detailed next. Note that this reverse engineering step may introduce systematic bias to the rest of learning as well if there is a mismatch between the actual ground truth. In some of our exploratory analysis on the mass inference results from BSP, we unexpectedly found that BSP can confuse heavy objects with light objects. This is different from the pattern of confusion that the human subjects display. We hypothesize this is due to the potential mismatch between the ground truth collision and friction forces and the reverse-engineered forces that we provide to BSP.

**Details for the learning task**  As there are three discs and three mats, the number of properties to infer is nine in total. The residual force to learn has the form: $C_1 \frac{f(q_1, q_2)}{\|\mathbf{P}_1 - \mathbf{P}_2\|_2^2} \boldsymbol{u} + C_2 \boldsymbol{u}_C$, where $C_1$ and $C_2$ are constants, $f$ itself is an expression of how the sign of the pairwise force depends on $q_i$ and $q_j$, $\boldsymbol{u}$ is the direction of the pairwise force and $\boldsymbol{u}_C$ is the direction of the global force (up, down, left or right).

**Questions and options presented to participants**    Participants are asked for a set of questions that would not reveal personally identifiable information.

1. Mass related questions (3)

   - How massive are [red] objects?
   - How massive are [yellow] objects?
   - How massive are [blue] objects?

Options are "Light", "Medium" and "Heavy".

2. Friction coefficient related questions (3)

   - How rough are [green] patches?
   - How rough are [purple] patches?
   - How rough are [brown] patches?

Options are "As smooth as the table-top", "A little rough" and "Very rough".

3. Pairwise force related questions (6)

   - How do [red] and [red] objects interact?
   - How do [red] and [yellow] objects interact?
   - How do [red] and [blue] objects interact?
   - How do [yellow] and [yellow] objects interact?
   - How do [yellow] and [blue] objects interact?
   - How do [blue] and [blue] objects interact?

Options are "Attract", "Repel" and "None".

4. Global force related questions (1)

   - Is a global force pulling the objects, and if so in what direction is it pulling?

Options are "Yes, it pulls North", "Yes, it pulls South", "Yes, it pulls East", "Yes, it pulls West" and "No global force".

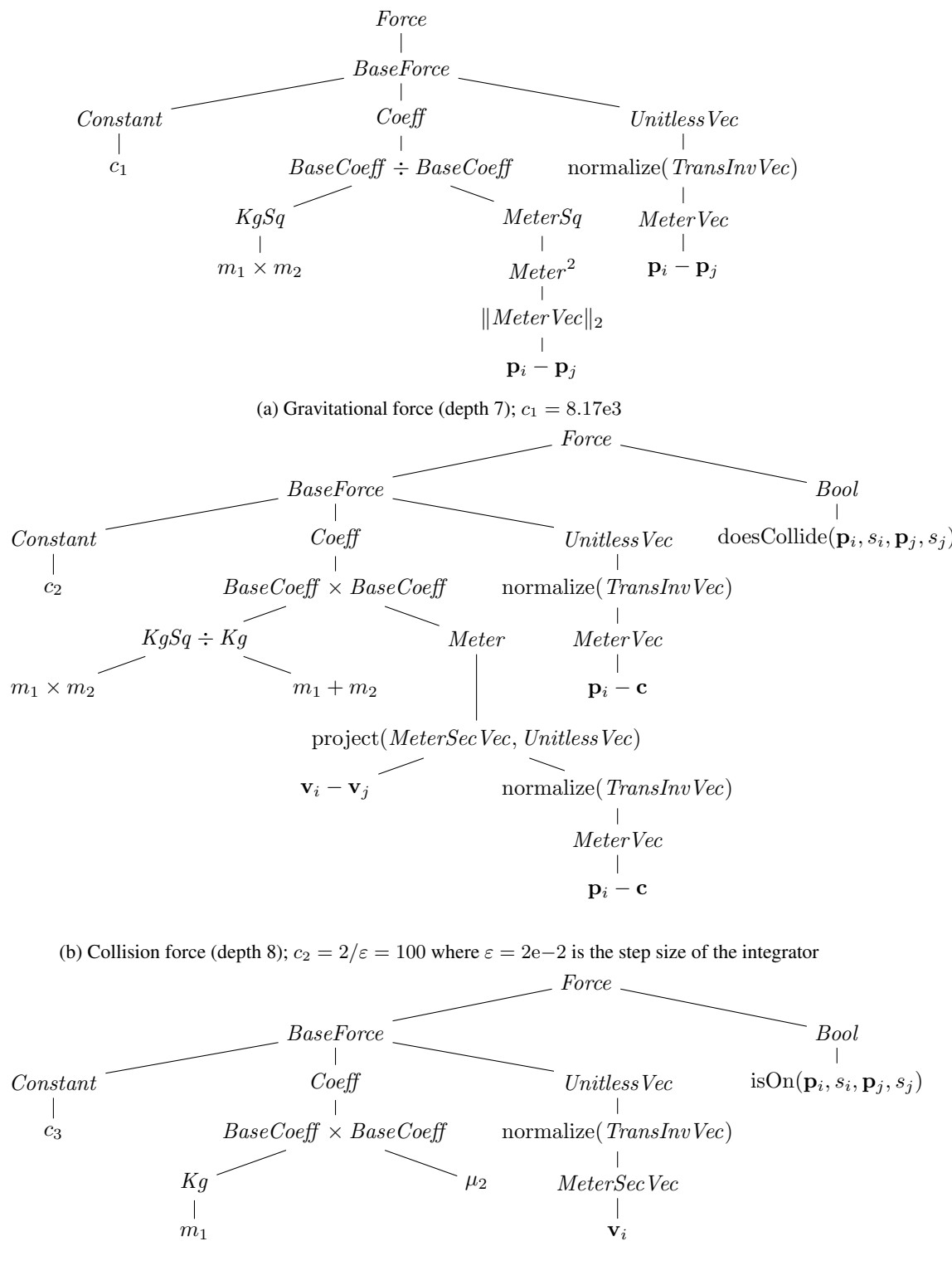

(a) Gravitational force (depth 7); $c_1 = 8.17\text{e}3$

(b) Collision force (depth 8); $c_2 = 2/\varepsilon = 100$ where $\varepsilon = 2\text{e}{-}2$ is the step size of the integrator

(c) Friction force (depth 5); $c_3 = 9.8$

Figure 12: Expression trees (under $\mathcal{G}$) of true force laws that generates the datasets used in section 4.

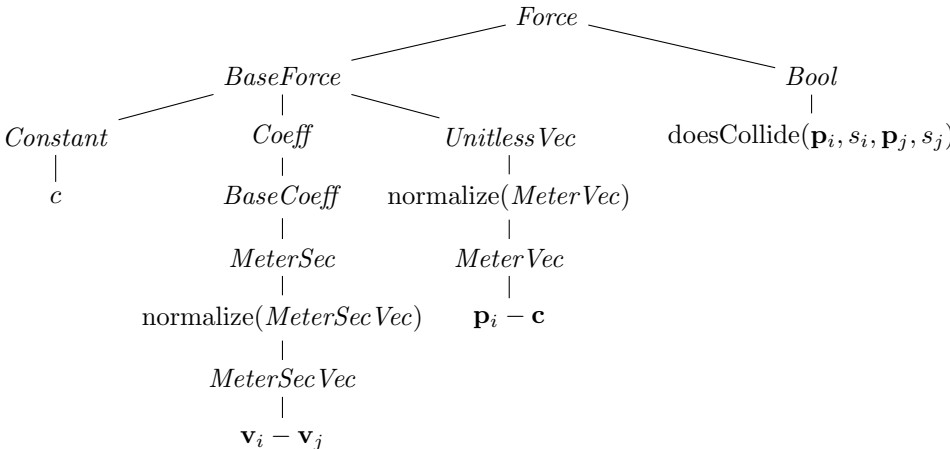

Figure 13: Approximate bounce law learned by BSP under our grammar; $c = 130.22$

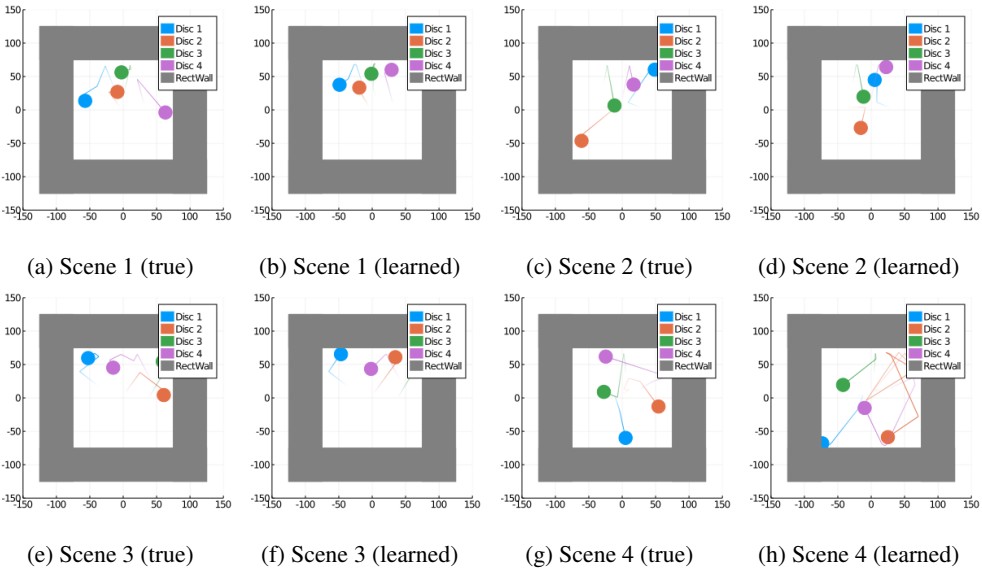

(a) Scene 1 (true)    (b) Scene 1 (learned)    (c) Scene 2 (true)    (d) Scene 2 (learned)

(e) Scene 3 (true)    (f) Scene 3 (learned)    (g) Scene 4 (true)    (h) Scene 4 (learned)

Figure 14: Predicated trajectories of the true bounce law and the learned bounce law.

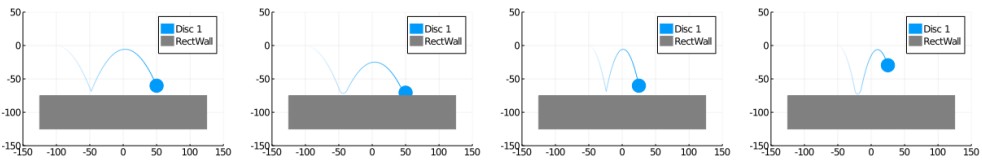

(a) Condition 1 (true)    (b) Condition 1 (learned)    (c) Condition 2 (true)    (d) Condition 2 (learned)

Figure 15: Generalization of the approximate bounce law in a vertical world with downward gravity.

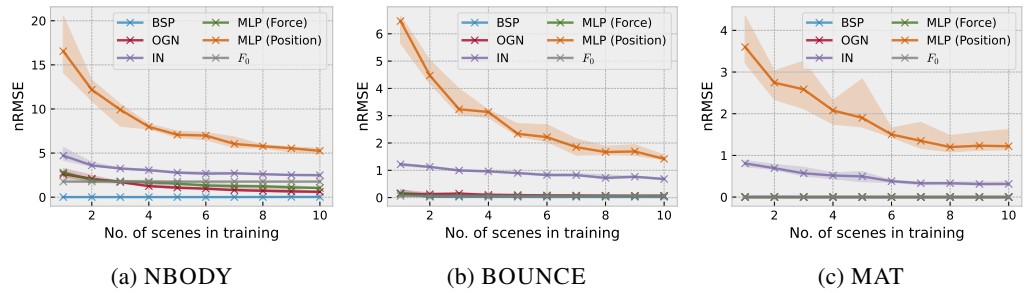

(a) NBODY      (b) BOUNCE      (c) MAT

Figure 16: Comparison of neural baselines and BSP, using predictive error on held out scenes given varying number of training scenes. Some baselines are not displayed due to very poor performance; see the appendix for version with all methods displayed.

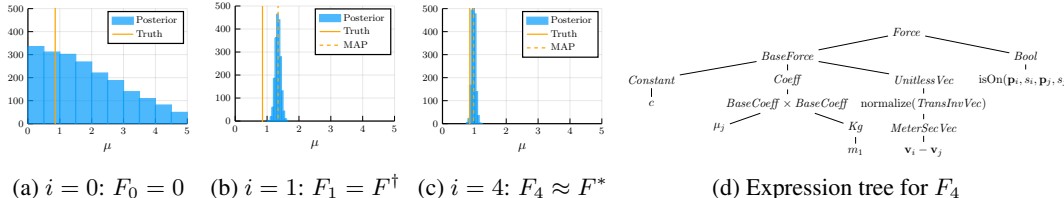

(a) $i = 0$: $F_0 = 0$    (b) $i = 1$: $F_1 = F^\dagger$    (c) $i = 4$: $F_4 \approx F^*$      (d) Expression tree for $F_4$

Figure 17: Results of the EM algorithm on MAT. figure 17a to figure 17c shows the posterior of friction coefficient in Scene 2 with the corresponding force function during EM. In figure 17b, the force function $F^\dagger = -22.99 \; \mu_j m_i \frac{\mathbf{v}_i}{\|\mathbf{v}_i\|_2}$ isOn$(\mathbf{p}_i, s_i, \mathbf{p}_j, s_j)$. The constant in figure 17d is $c = -8.605$.

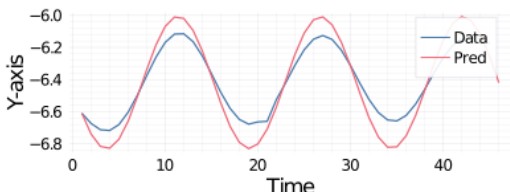

Figure 18: Prediction of the vertical position