# OpenReview forum: "A Bayesian-Symbolic Approach to Reasoning and Learning in Intuitive Physics"
_NeurIPS.cc/2021/Conference — NeurIPS 2021 Poster_

### Official Review · Reviewer_6f6C · 2021-07-14

**Rating:** 6
**Confidence:** 4

**Summary:**

The authors propose the Bayesian symbolic (BSP) framework for inferring both the symbolic force laws and the values of latent properties of entities.
They contribute a grammar of Newtonian physics over which they perform inference with the BSP framework.
Empirical results show higher sample efficiency than neural approaches, and performance that is between humans and chance on the ULLMAN dataset.

**Ethical Concerns:**

No issues.

**Limitations And Societal Impact:**

Yes, they were addressed.

**Main Review:**

Originality: on the one hand, the method can be seem as novel compared to the approaches in Figure 1. On the other hand, the methodology seems to be straightforward variant of [0] to intuitive physics domains.

Clarity: the writing and presentation is clear.

Quality:

While it is true that neural approaches are more sample inefficient, Figure 4 is also an unfair comparison. As the authors acknowledge in section 6, they assume that BSP has access to a grammar of Newtonian physics that neural approaches do not have. Indeed, neural approaches must both learn the grammar and how the grammar applies to a particular dataset, while BSP only needs to learn about the latter. Therefore, the results in Figure 4 do not seem to be very informative of possible advantages BSP has over neural baselines.

What might make for a better empirical case for BSP is if the authors could dig deeper into Figure 1 and identify explicit axes along which to measure the advantages and disadvantages of BSP vs other baselines in transfer to a new domain *after* the model has been trained. This could help address the problem with the unequal starting point between BSP and neural approaches. For example,
- suppose we first learn in a multi-task fashion on BOUNCE and MAT, and then transfer to a domain with balls bouncing around over a mat? How would this compare with IN [1] or NPE [2]?
- the authors state that [tracking algorithm + BSP] learns Hooke's law, after having trained on this environment, what would be the sample efficiency of [tracking algorithm + BSP] be on a spring environment with a different spring constant? And how would this compare with a neural approach that has object perception built in, such as [3]?

Can BSP generalize to completely different kinds of domains from that which it has been trained, such as generalizing to collisions of cars in a simulated freeway from training on the BOUNCING environment? Presumably that would be what I'd expect the strength of symbolic regression to be, it would strengthen the paper if this hypothesis were true, or alternatively if the authors could provide an explanation of why this hypothesis is false.

Significance:

I am assuming that the authors propose BSP and the Newtonian grammar as a hypothesis for a computational model of intuitive physics in humans.
In this case it would have been enlightening to see how BSP compares to the computational model in Ullman et al. (2018) -- would the authors be able to provide this comparison, and comment on the tradeoffs between the two approaches.
Without a proper comparison with an alternative hypothesis (e.g. Ullman et al. 2018) as a baseline, rather than simply the random chance baseline, it is not clear how to judge the merits of BSP in modeling humans.

Questions:

How does the wall clock time of inference compare between humans, BSP, and a neural method like [1] or [2]?

I was not able to find the appendix, but I think it would be very valuable to see to understand the proposed method.

Summary:
Overall, while the proposed idea is intriguing, the empirical results seem to fall slightly short of clearly elucidating the merits of the method. The comparison to neural baselines would likely need to be in a transfer setting in order to be a fair comparison, and the human experiments require a discussion with an additional baseline (e.g. Ullman et al. 2018) on the axes along which one or the other method is a better explanation for human intuitive physics induction.

[0] Ellis, Kevin, Catherine Wong, Maxwell Nye, Mathias Sable-Meyer, Luc Cary, Lucas Morales, Luke Hewitt, Armando Solar-Lezama, and Joshua B. Tenenbaum. "Dreamcoder: Growing generalizable, interpretable knowledge with wake-sleep bayesian program learning." arXiv preprint arXiv:2006.08381 (2020).
[1] Battaglia, P. W., Pascanu, R., Lai, M., Rezende, D., & Kavukcuoglu, K. (2016). Interaction networks for learning about objects, relations and physics. arXiv preprint arXiv:1612.00222.
[2] Chang, M. B., Ullman, T., Torralba, A., & Tenenbaum, J. B. (2016). A compositional object-based approach to learning physical dynamics. arXiv preprint arXiv:1612.00341.
[3] Veerapaneni, R., Co-Reyes, J. D., Chang, M., Janner, M., Finn, C., Wu, J., ... & Levine, S. (2020, May). Entity abstraction in visual model-based reinforcement learning. In Conference on Robot Learning (pp. 1439-1456). PMLR.



**Time Spent Reviewing:**

1.5

---

> ### Author Response · Authors · 2021-08-10
> **Separate response**
>
> Thank you a lot for your comments. We address some of the concerns in the common response and please find below to some of your special remarks:
>
> > the authors state that [tracking algorithm + BSP] learns Hooke's law, after having trained on this environment, what would be the sample efficiency of [tracking algorithm + BSP] be on a spring environment with a different spring constant?
>
> In the setup you describe, BSP only needs to re-learn the global constant, which was fit by the inner-level optimization in the first training. In cases where the force law is correctly learned, we find that BSP can always learn the correct spring constant with only one scene.
>
> > Can BSP generalize to completely different kinds of domains from that which it has been trained, such as generalizing to collisions of cars in a simulated freeway from training on the BOUNCING environment?
>
> Yes, it can. As proof of concepts, we have done experiments of learning the force from BOUNCE, where we have a top-down view of the world, and testing the learn force in a ball throwing scene where we have a front view of the world (i.e. the ball has a vertical motion component). This is available in appendix B.3.
>
> > In this case it would have been enlightening to see how BSP compares to the computational model in Ullman et al. (2018) -- would the authors be able to provide this comparison, and comment on the tradeoffs between the two approaches.
>
> The reason we didn't include the computational model in Ullman et al. (2018) is that the forms of force functions to learn are very different. The task there is to infer binary variables to activate some predefined force components. In the context of BSP, this would become an inference only setup with binary variables for predefined force components. We argue that such learning tasks are very different and not directly comparable. see the discussion in the paper (L340-342).
>
> > The comparison to neural baselines would likely need to be in a transfer setting in order to be a fair comparison.
>
> We respectfully disagree.  Our baselines are specifically chosen to support our scientific claim that our method for adding domain knowledge improves on SOTA methods that do not use more domain knowledge. Please refer to our common response for more comprehensive discussion.

---

> > ### Comment · Reviewer_6f6C · 2021-08-28
> > **Response to Rebuttal**
> >
> > Thanks for your response.
> >
> > I still have concerns about the comparison to the neural baselines, which I still believe is unfair.
> >
> > The authors state "Our baselines are specifically chosen to support our scientific claim that our method for adding domain knowledge improves on SOTA methods that do not use more domain knowledge." If the claim is that "adding more domain knowledge will improve performance," I'm not sure this claim is very surprising: information that does not come from the data must come from the prior, so if both the proposed method and neural baselines receive the same data, but the proposed method has more prior knowledge than the neural baselines, wouldn't we expect the proposed method to perform better? If the neural baselines must both learn the grammar and how the grammar applies to a particular dataset, while BSP only needs to learn about the latter, and both approaches are given the same amount of data, then of course BSP would be better. Taken further, suppose one were to augment the grammar with Hooke's law as a primitive - I might expect this augmented method to have better sample efficiency than BSP. Simply showing that a method with more prior knowledge performs better than a baseline that does not have any means to acquire that prior knowledge seems like a strawman argument.
> >
> > This is why I suggested a comparison in a transfer setting: if we allow the neural baseline and BSP to train on one set of tasks, and then measure their sample efficiency in learning on a different set of tasks that also exhibit similar laws of physics (but might have different objects or scene configurations), then at least the neural baseline has had enough data to learn to model the laws in the first place. Simply giving the neural baseline only 10 scenes (Fig 4) does not seem nearly enough, especially when we consider that human priors have been learned over thousands of years of evolution.
> >
> > I suppose a less critical interpretation of the claim that "adding domain knowledge improves on SOTA methods that do not use more domain knowledge" is that perhaps the authors are making a claim that this kind of domain knowledge (in the form of the BSP grammar) might be a better theory for explaining human predictions than a neural baseline? In that case, it would be useful to have a fair comparison in Table 3 BSP with an Interaction Network or Neural Physics Engine, which has had enough training data to be able to make reasonable predictions on the training set, and see if these neural baselines can actually generalize to novel scenes in the way BSP can.

---

> > > ### Author Response · Authors · 2021-08-31
> > > **Response to comments**
> > >
> > > > Thanks for your response.
> > >
> > > We thank the reviewer for getting back to us. We provide clarification to their doubts below.
> > >
> > > > I still have concerns about the comparison to the neural baselines, which I still believe is unfair.
> > > The authors state "Our baselines are specifically chosen to support our scientific claim that our method for adding domain knowledge improves on SOTA methods that do not use more domain knowledge." If the claim is that "adding more domain knowledge will improve performance," I'm not sure this claim is very surprising: information that does not come from the data must come from the prior, so if both the proposed method and neural baselines receive the same data, but the proposed method has more prior knowledge than the neural baselines, wouldn't we expect the proposed method to perform better?
> > >
> > > As the reviewer correctly points out, comparisons to neural network based models of intuitive physics are to establish that methods that allow for adding domain knowledge, perform better that those that don’t on the task of intuitive physics, which has not been demonstreated before. However, we would also like to clarify these experiments are also to demonstrate that our main contribution, that is, the BSP method, is the only method that allows for adding domain knowledge with human-level sample efficiency in comparison to the other models of intuitive physics. This is clearly not an obvious result, and therefore we establish it by comparing BSP to the SOTA models of intuitive physics.
> > >
> > > > If the neural baselines must both learn the grammar and how the grammar applies to a particular dataset, while BSP only needs to learn about the latter, and both approaches are given the same amount of data, then of course BSP would be better. Taken further, suppose one were to augment the grammar with Hooke's law as a primitive - I might expect this augmented method to have better sample efficiency than BSP. Simply showing that a method with more prior knowledge performs better than a baseline that does not have any means to acquire that prior knowledge seems like a strawman argument.
> > >
> > > For neural baselines, the networks do not need to learn a grammar in order to complete the task. Following the terminology the reviewer uses here, the neural network only needs to learn the final expression (of physics laws) for a particular dataset, but not the grammar, to complete the task. The grammar we provide to BSP does not encode knowledge like "Hooke's law as a primitive" either. As discussed, the two priors provided are generically true for various physical laws.
> > >
> > > > This is why I suggested a comparison in a transfer setting: if we allow the neural baseline and BSP to train on one set of tasks, and then measure their sample efficiency in learning on a different set of tasks that also exhibit similar laws of physics (but might have different objects or scene configurations), then at least the neural baseline has had enough data to learn to model the laws in the first place. Simply giving the neural baseline only 10 scenes (Fig 4) does not seem nearly enough, especially when we consider that human priors have been learned over thousands of years of evolution.
> > > > I suppose a less critical interpretation of the claim that "adding domain knowledge improves on SOTA methods that do not use more domain knowledge" is that perhaps the authors are making a claim that this kind of domain knowledge (in the form of the BSP grammar) might be a better theory for explaining human predictions than a neural baseline? In that case, it would be useful to have a fair comparison in Table 3 BSP with an Interaction Network or Neural Physics Engine, which has had enough training data to be able to make reasonable predictions on the training set, and see if these neural baselines can actually generalize to novel scenes in the way BSP can.
> > >
> > > Since our current experiments clearly establish our objective i.e. “the BSP model of intuitive physics reaches human-level performance by allowing for the  incorporation of domain knowledge in a sample efficient manner”, we do not think the suggested transfer learning tasks will add any further insight to this objective. This is because even if the neural-network based baselines perform comparably to BSP in the transfer task, they will require far more data, hence will not be sample efficient.
> > >
> > > Further, it is not clear how to design these experiments for neural-network based models such that they chose to learn the common priors of physics instead of simply optimizing for the tasks they are trained on. To be more specific, in a transfer learning setup, it is unclear what types of knowledge would be learned in the first stage - whether it is similar to the generic physics priors we are discussing or something more task-specific to the scenes (such as knowledge about the types of common objects, the shared environments, etc). In short, neural-network based models of intuitive physics do not have a clear way to add domain knowledge.

---

> > > > ### Comment · Reviewer_6f6C · 2021-09-01
> > > > **Response**
> > > >
> > > > > Since our current experiments clearly establish our objective i.e. “the BSP model of intuitive physics reaches human-level performance by allowing for the incorporation of domain knowledge in a sample efficient manner”, we do not think the suggested transfer learning tasks will add any further insight to this objective. This is because even if the neural-network based baselines perform comparably to BSP in the transfer task, they will require far more data, hence will not be sample efficient.
> > > >
> > > > > Further, it is not clear how to design these experiments for neural-network based models such that they chose to learn the common priors of physics instead of simply optimizing for the tasks they are trained on. To be more specific, in a transfer learning setup, it is unclear what types of knowledge would be learned in the first stage - whether it is similar to the generic physics priors we are discussing or something more task-specific to the scenes (such as knowledge about the types of common objects, the shared environments, etc). In short, neural-network based models of intuitive physics do not have a clear way to add domain knowledge.
> > > >
> > > > The fair comparison I was thinking of for transfer would be the following.
> > > > (1) train the neural baseline on tasks that would require it to learn the same physics priors encoded in the BSP grammar. Unlimited data would be allowed here because humans had a lot of data from evolution to come up with the physics priors encoded in the BSP grammar.
> > > > (2) compare the pre-trained neural baseline and the BSP model on a new task (it could be the tasks given in this paper), and compare the neural baseline and the BSP model's sample efficiency on this new task.
> > > >
> > > > Concretely, for example, we could consider pre-training the neural baseline on various different masses in the BOUNCE environment, such that it learns the physics prior of "collision."  This would be (step 1) above. Then the held out task (step 2) would be to test sample efficiency on a prediction task with a different set of masses than seen during training (or the task in step 2 could be generalization in some other way, such as generalizing to more objects or to a different elasticity coefficient). This way, the neural baseline will have learned the prior knowledge that "objects interact via forces" (which is baked into the BSP grammar), and then we would have a fair comparison in step 2 on each methods' sample efficiency in inferring the *specific* force for a particular scene, given that prior knowledge.
> > > >
> > > > I would see this as a fairer comparison because we'd be comparing sample efficiency on learning a task *after* the physics priors have been encoded: for BSP the prior was encoded by the human via the grammar, and for the neural baseline the prior would be encoded via pre-training.

---

> > > > > ### Author Response · Authors · 2021-09-01
> > > > > **Response**
> > > > >
> > > > > > The fair comparison I was thinking of for transfer would be the following. (1) train the neural baseline on tasks that would require it to learn the same physics priors encoded in the BSP grammar. Unlimited data would be allowed here because humans had a lot of data from evolution to come up with the physics priors encoded in the BSP grammar. (2) compare the pre-trained neural baseline and the BSP model on a new task (it could be the tasks given in this paper), and compare the neural baseline and the BSP model's sample efficiency on this new task.
> > > > >
> > > > > > Concretely, for example, we could consider pre-training the neural baseline on various different masses in the BOUNCE environment, such that it learns the physics prior of "collision." This would be (step 1) above. Then the held out task (step 2) would be to test sample efficiency on a prediction task with a different set of masses than seen during training (or the task in step 2 could be generalization in some other way, such as generalizing to more objects or to a different elasticity coefficient). This way, the neural baseline will have learned the prior knowledge that "objects interact via forces" (which is baked into the BSP grammar), and then we would have a fair comparison in step 2 on each methods' sample efficiency in inferring the _specific_ force for a particular scene, given that prior knowledge.
> > > > >
> > > > > For the setting described above, in step 1, the model can learn both the physics prior as well as the exact force law that is needed for the scenes in step 2. It’s unclear what is to transfer here and why the training in step 2 is needed. In fact, training on different masses only makes sense if the model overfits to the data in step 1; training on different number of objects is not needed for the IN and OGN baselines because they are designed to generalise to different number of objects by construction.
> > > > >
> > > > > Regarding including a neural baseline that has the prior knowledge that "objects interact via forces”, the OGN baseline serves exactly this purpose. To be more concrete, the only neural component to learn in the OGN model is a function that takes symbolic inputs from a pair of objects and outputs “force” between them. These outputs for each object is then summed up (as forces are additive). Denote all summed forces applied to all objects as $f$ (which is the force applied to the system), $f$ is then used in the Euler integrator as the force to update the position by the following steps
> > > > > $$
> > > > > a = f / m, v = v + a \Delta t, \quad p = p + v \Delta t,
> > > > > $$
> > > > > where $a$ is for acceleration, $m$ is for mass, $v$ is for velocity, $p$ is for position and $\Delta t$ is the step size of the integrator; see equation (1) for a version with time indexing.
> > > > > In fact, because OGN has this prior that other neural baselines do not have (or only partially have), OGN is the most competitive model in two out of three scenes in figure 4.

---

> > > > > > ### Comment · Reviewer_6f6C · 2021-09-03
> > > > > > **Response**
> > > > > >
> > > > > > I understand more about the difficulty in providing a fair comparison between the neural baselines and the proposed method. I have updated my score to a 6.

---

### Official Review · Reviewer_9Hgp · 2021-07-14

**Rating:** 6
**Confidence:** 4

**Summary:**

This paper is proposing a new Bayesian symbolic framework that can assist in being able to perform types of physical reasoning over items and environments. The motivation in the paper is that humans are able to understand the physical properties of a system given very few data samples therefore we should be able to develop machine learning systems that can determine the underlying physical properties of the environment with similar data efficiency. The idea in the paper is to use a Bayesian inference method in order to be able to determine a list of parameters and how to combined different functions to describe the physics of a particular environment.

Pros
- The method does propose an interesting grammar that can be used to more efficiently explore the search space of possible physical functions to describe an environment
- The method also shows some promising performance across a number of different environments for understanding the underlying physical properties and only a few data samples.
- Learning the physical functions as a combination of terms is also a promising direction for this research. It could be interesting to explore this further and apply the method to systems where the physics may not be well known to discover the physical properties of new physical systems.

Cons
- The assumptions for the model are still not perfectly clear and well contextualized with respect to prior work. This is referring to where and how the symbolic representation for the items in the environment comes from. It's not clear if prior methods also have access to this or similar information.
- It would also be good to better outline the limitation on the types of physical systems this model can learn to predict over. The equations that are supported seem to indicate the most Newtonian/classical physical systems should be possible but it is not clear what are the limitations of the grammar, along with the types of functions that can be represented.
- Well it is mentioned in section 6, it's not clear how the perceptual video data used in the real experiments or The ULLMAN experiment is processed into symbolic values that can be used by BSP. Is this mechanism easy to automate such that it's not a significant limitation of applying the method in the paper?



**Limitations And Societal Impact:**

Well it is mentioned in section 6, it's not clear how the perceptual video data used in the real experiments or The ULLMAN experiment is processed into symbolic values that can be used by BSP. Is this mechanism easy to automate such that it's not a significant limitation of applying the method in the paper?

**Main Review:**

Additional comments on the work:
- It says the environment is model is a top-down generative model in which things are assumed to interact with each other, does this mean that the method requires the top-down model beforehand?
- Figure one in the paper is challenging to understand. Is each column designed to represent the different papers that are described in the caption? If this is true then this figure is displaying a lot of information but it is difficult to understand the connections between these different papers and what is the significance of the transitions between them. It is important to discuss this well in the related work section to help contextualize how this work fits into prior methods and how it is different.
- In the introduction, it's outlined that the paper introduces a grammar for Newtonian physics that also greatly helps the overall learning problem. It would be helpful to be able to describe this grammar more in the introduction so the reader can already get an intuition for why this grammar is helpful.
- From the description given in section 3.1 the model is given a vector of intrinsic physical properties for each of the items in the environment. This particular information can greatly assist in being able to learn the overall problem and being able to estimate interacting bodies between them. It would be good to know if previous methods use a similar structure in this paper or if the primary contribution is being able to take advantage of this type of structure where other methods have not had access to this graph of intrinsic properties.
- While the first feature of the grammar described in the paper makes sense such that units are preserved between operations. The second point of making sure to use absolute positions feels like something that is well known in the machine learning community as it reduces the complexity of the problem in order to learn behaviors that are more egocentric and less absolute.
- In section 4.1 the paper reminds the reader that the average person can perform physical learning tasks similar to the ones we use with fewer than five scenes. Yet it seems that the environments described in the paper are new and have not been evaluated on people to understand how many scenes are needed for a person to be able to accurately understand and perform predictions in those physical simulations. It is important to better validate these environments to make sure that humans can easily understand the physics in them given just a few scenes. This will also provide information with respect to the motivation that people can understand these systems very quickly. On the other hand there may be environments that already exist in the community that can be used to perform these types of experiments. For example, the environments used in the Battaglia at all, 2016 paper.
- A number of the referenced papers are citing ArXiv versions of the papers when the papers are already published. The proper published papers should be cited in the paper not the ArXiv versions. For example, Interaction networks for learning about objects, relations and physics.
- The ablations in section 4.2 are interesting and also provide good information as to the helpfulness of the grammar that's designed in the paper. However part of the motivation for creating the grammar was to make the learning problem more tractable. Therefore it would also be helpful to include an additional comparison over the same ablations but when more time is given to the training algorithm in order to learn the problem. In this case, the method proposed in the paper should be able to learn quickly with fewer iterations of learning in order to be able to discover the proper physical settings while not using the full grammar may be able to learn with the same performance but will require a significant amount of time that is undesirable.
- For the analysis and section 5.2 how does bsp compare to the prior neural-based methods for being able to do predictions in those real environments? It's not mentioned whether or not those real environments are impossible to solve using the neural-based methods. Some additional context here in comparison to prior work would be extremely helpful to the reader and help the reader understand how the method performs outside of the SYNTH environments that were created for this paper.
- Well it is mentioned in section 6, it's not clear how the perceptual video data used in the real experiments or The ULLMAN experiment is processed into symbolic values that can be used by BSP. Is this mechanism easy to automate such that it's not a significant limitation of applying the method in the paper?

----- Post Author Response ----
I have updated my score thanks to the authors' responses and the supplementary material.

**Time Spent Reviewing:**

2

---

> ### Author Response · Authors · 2021-08-10
> **Separate response**
>
> Thank you a lot for your comments. We address some of the concerns in the common response and please find below to some of your special remarks:
>
> > The assumptions for the model are still not perfectly clear and well contextualized with respect to prior work. This is referring to where and how the symbolic representation for the items in the environment comes from. It's not clear if prior methods also have access to this or similar information.
>
> While some prior works assume direct access to symbolic inputs [1,2] others train a “de-render” to convert perception data into symbolic forms [3,4]. In this work, we provide both our method and the baselines we compare against the full symbolic representation. This makes the comparison fair and accurate. We didn't put much effort into developing a perceptual model because our focus is on the priors for force learning instead of parsing scenes. Therefore, for scenes in which symbolic representations are not directly accessible, we “de-render” the visual scene into symbolic representation in a separate pre-processing step. We will make it clear in our revised draft.
>
> [1] Battaglia, P. W., Pascanu, R., Lai, M., Rezende, D., and Kavukcuoglu, K. Interaction networks for learning about objects, relations and physics, 2016
> [2] Cranmer, M., Sanchez-Gonzalez, A., Battaglia, P., Xu, R., Cranmer, K., Spergel, D., and Ho, S. Discovering symbolic models from deep learning with inductive biases, 2020
> [3] Wu, J., Yildirim, I., Lim, J. J., Freeman, B., and Tenenbaum, J. Galileo: Perceiving physical object properties by integrating a physics engine with deep learning, 2015
> [4] Smith, K., Mei, L., Yao, S., Wu, J., Spelke, E., Tenenbaum, J., and Ullman, T. Modeling expectation violation in intuitive physics with coarse probabilistic object representations, 2020
>
> > Well it is mentioned in section 6, it's not clear how the perceptual video data used in the real experiments or The ULLMAN experiment is processed into symbolic values that can be used by BSP. Is this mechanism easy to automate such that it's not a significant limitation of applying the method in the paper?
>
> We've included the missing appendix via this anonymous link https://drive.google.com/drive/folders/13BZvuj0BBOF7pg2XrYGurSB-cdhzki4o as suggested by PCs, in which appendices C.2 and C.3 describe the pre-processing steps. Basically either some off-the-shelf tracking algorithms can be used, or some self-supervised de-render can be trained and used as the perception module.
>
> > It says the environment is model is a top-down generative model in which things are assumed to interact with each other, does this mean that the method requires the top-down model beforehand?
>
> Yes, this is correct. To clarify, while we represent the top-down model for each scene as a probabilistic program, BSP acts on these abstraction of probabilistic programs directly without the need of being implemented for each scene. One only needs to specify the types of entities and the properties to infer in each program and the rest is automated.
>
> > Figure one in the paper is challenging to understand. Is each column designed to represent the different papers that are described in the caption?
>
> Yes, each column is a type of approach and at least one paper implements that idea, as we discussed in section 2.
>
> > From the description given in section 3.1 the model is given a vector of intrinsic physical properties for each of the items in the environment. This particular information can greatly assist in being able to learn the overall problem and being able to estimate interacting bodies between them. It would be good to know if previous methods use a similar structure in this paper or if the primary contribution is being able to take advantage of this type of structure where other methods have not had access to this graph of intrinsic properties.
>
> Methods in figure 1 either directly take symbolic inputs or use derenderers to obtain symbolic representations, and they all assume fully observed intrinsic properties (i.e. symbolic representations). Only BSP allows for maintaining uncertainty over intrinsic properties by treating them as unobserved random variables. This is in fact one of the key technical novelties of our model.
>
> > The second point of making sure to use absolute positions feels like something that is well known in the machine learning community as it reduces the complexity of the problem in order to learn behaviors that are more egocentric and less absolute.
>
> Yes, you are right and for fair comparison, we indeed encode this knowledge into our implementation of INs and OGNs, which is what the original papers do as well.
>
> > On the other hand there may be environments that already exist in the community that can be used to perform these types of experiments. For example, the environments used in the Battaglia at all, 2016 paper.
>
> We design our SYNTH environment as a "break-down" of ULLMAN - where each scene corresponds to one force law in ULLMAN. Also, the environments used in the Battaglia et al. (2016) are not too different from SYNTH as both contain NBODY and BOUNCE.
>
> > For the analysis and section 5.2 how does bsp compare to the prior neural-based methods for being able to do predictions in those real environments? It's not mentioned whether or not those real environments are impossible to solve using the neural-based methods.
>
> As the neural-based methods were not originally introduced in a context where the properties are unknown, the most straightforward way to use them is as the M-step in BSP. However, we were not able to achieve sensible results with the very limited number of training scenes for neural baselines. This can also be observed  by the unsatisfying performance of neural baselines for <= scenes in figure 4. Note their performance is even worse than the zero force (constant velocity) baseline.

---

### Official Review · Reviewer_9SzY · 2021-07-15

**Rating:** 6
**Confidence:** 3

**Summary:**

This paper presents the Bayesian-symbolic framework (BSP), a novel framework that combines symbolic reasoning and learning to discover intuitive physics. BSP uses learning to infer object properties, and uses symbolic expression to recover physical laws. The models are learned in an EM manner, where BSP alternates between inferring the object properties with the current force, and inferring the forces given the object property distributions. The dynamics are assumed to be Newtonian. The experiments show that BSP outperforms selected baselines, and achieve relatively good results compared to humans on the SYNTH benchmark and PHYS101 dataset.


**Limitations And Societal Impact:**

Yes, limitations and impact are discussed in section 6.

**Main Review:**

=== Strengths ===

+ The paper is well written and the presentation is strong.

+ The evaluations on physical forces in section 5.1 are comprehensive, as it covers both gravity, collision, and friction.

+ The general idea of combining symbolic reasoning and statistical inference is nice. It also bypasses the non-differentiability issue in collisions, often encountered by pure neural network methods.

+ The use of prior knowledge such as dimensional analysis to prune the symbolic search is neat, as these are also tricks that people commonly use in physics.

=== Weaknesses ===

- This is my opinion only, but the EM optimization method seems slightly naive here. How often would such a search get stuck in the local optimum or saddle points, for example, due to an wrongly searched expression?

- I am really curious to know whether BSP can discover certain physical constants. For example, the Kepler constant K of Earth, or the gravitational constant G? This is my opinion only, but these constants are more exciting compared to the properties such as masses or the spring constants analyzed in section 5.

- I am slightly concerned about the performance of BSP in table 3, as it does not outperform humans. I would imagine humans are not good at such tasks, and would expect better results from BSP, especially considering the task is more of a classification problem rather than inferring the exact physical quantity. Could the authors explain this more in detail?


**Time Spent Reviewing:**

5

---

> ### Author Response · Authors · 2021-08-10
> **Separate response**
>
> Thank you a lot for your comments. We address some of the concerns in the common response and please find below to some of your special remarks:
>
> > This is my opinion only, but the EM optimization method seems slightly naive here. How often would such a search get stuck in the local optimum or saddle points, for example, due to an wrongly searched expression?
>
> Getting stuck in local optima is a fundamental issue in symbolic search and no method can entirely avoid it: BSP currently relies on random initialisation to tackle it, as is common in EM-based approaches. In addition, one of our key technical contributions -- the bilevel optimisation in the M-step -- is based on the goal of lessening the problem of local minima: If we estimated the constants during the E-step, then it would be very hard to change to a new formula that worked for those constants. We found such a specialised M-step together with random initialisation works well for the tasks we consider in our paper.
>
> > I am really curious to know whether BSP can discover certain physical constants. For example, the Kepler constant K of Earth, or the gravitational constant G?
>
> Yes! As mentioned in the **Novelty** part of our common response, BSP indeed recovers the global constants in all three scenes from SYNTH (gravitational constant G, gravitational constant g and bounce coefficient) as well as the gravitational constant g in FALL. This is one of the main technical strengths of BSP: Other alternatives can either learn these quantities implicitly or cannot model them at all.
>
> > I am slightly concerned about the performance of BSP in table 3, as it does not outperform humans. I would imagine humans are not good at such tasks, and would expect better results from BSP, especially considering the task is more of a classification problem rather than inferring the exact physical quantity. Could the authors explain this more in detail?
>
> We posit that the better sample efficiency of humans compared to BSP can be attributed to the fact that humans have a large amount of experience with real-world physics. Although the physical laws in ULLMAN are not real-world physical laws, they share enough similarities that we expect people can often learn them quickly based on their past experience.

---

### Official Review · Reviewer_2v2s · 2021-07-17

**Rating:** 6
**Confidence:** 4

**Summary:**

A neural-symbolic Bayesian framework for learning intuitive physical models, by making abduction from a few observations provided by the environment and domain knowledge given by grammar.

**Limitations And Societal Impact:**

* **Novelty**: Although exploiting the E-M algorithm and S-R method for solving reasoning-by-observation problems like the ones presented in this work is valid, this method has already been an established and typical framework for explanatory learning that tries to learn a (logical and statistical) model from raw observation and ungrounded primitives. Hence, I think the proposed method is not a principle contribution for novelty. A domain-specific PCFG is a canonical workstream in the neural-symbolic community, either. The novelty of this work comes in applying the learning-by-explanation framework to induce physical rules with data efficiency. This is a principle contributor to the community, yet it goes not so far as the authors claimed;

* **Scalability**: The authors exploit heuristics like checking the consistency of measurement for pruning the hypothesis space, which serves as metaknowledge. This is a good trial but make the method much too ad-hoc. It may be better if formulated in a formal logic way, making the symbols generalizable. Either, the given grammar is way overstrong, making both the framework hard to extrapolate and the task trivial. It would be better if the framework makes abstraction over a small set of primitives;

* **Validity**: Since physical reasoning is not entirely a novel task for the community, it is necessary to carefully benchmark methods for the task. However, the authors only compared with four end-to-end learning methods, without consideration of neural-symbolic counterparts like DeepProblog and Abductive Learning. Since neural methods can hardly encode domain knowledge, the comparison is naturally unfair. It may make the results more solid if the authors compare their methods with neural-symbolic counterparts using identical domain knowledge.

**Main Review:**

This paper proposes a neural-symbolic Bayesian framework for learning intuitive physical models, by making abduction from a few observations provided by the environment and domain knowledge given by grammar. The framework is consistent with the general methodology, i.e., *sample(hypothesis)-search-observation loop* of learning-by-explanation paradigm. The solutions, E-M algorithm, and S-R are also canonical methods for combining symbolic knowledge with statistical observation, where the former serves as the hypothesis-generator operation and the latter selects a most plausible hypothesis as the interpretation of the observation. In general, the authors formulate a significant problem and provided the community with a human-like computational approach.

**Time Spent Reviewing:**

2

---

> ### Author Response · Authors · 2021-08-10
> **Separate response**
>
> We thank the reviewer for the feedback and comments. We address some of the concerns in the common response and please find below to your special remark:
>
> > The authors exploit heuristics like checking the consistency of measurement for pruning the hypothesis space, which serves as metaknowledge. This is a good trial but makes the method much too ad-hoc.
>
> We would argue that the grammar is not ad-hoc because the design of the grammar is based on two general principles that we expect to hold across a largest of physical settings: The consistency check is based on dimensional analysis and the removal of absolute position is motivated to introduce Galilean invariance. Several prior works such as [1] have made use of these priors in a different context.
>
> [1] Udrescu, S.-M. and Tegmark, M. Ai feynman: A physics-inspired method for symbolic regression. 2020

---

### Decision · Program_Chairs · 2021-09-27

**Decision:**

Accept (Poster)

**Comment:**

The paper presents a neural-symbolic Bayesian framework to learning physical models in a similar way as human, by making abduction from a few observations provided by the environment and domain knowledge given by grammar. It aims to address an important problem. The proposed solution is reasonable with promising results.

A few concerns have been raised: (1) evaluation: stronger baselines should be included and some results are not as good as human; (2) how the grammar was learned - ad hoc versus principled ways; (3) how general the models can be - only for some specific physical systems versus general systems. The authors were able to address these concerns to a certain extent. After discussion, it was agreed that while the paper still has some limitations, the work is interesting enough to be presented to the NeurIPS community.